

# Diurnal variability, photochemical production and loss processes of hydrogen peroxide in the boundary layer over Europe

Horst Fischer[1], Raoul Axinte[1], Heiko Bozem[1*], John N. Crowley[1], Cheryl Ernest[1**], Stefan Gilge[2], Sascha Hafermann[1], Hartwig Harder[1], Korbinian Hens[1], Rainer Königstedt[1], Dagmar Kubistin[1,2], Chinmay Mallik[1], Monica Martinez[1], Anna Novelli[1#], Uwe Parchatka[1], Christian Plass-Dülmer[2], Andrea Pozzer[1], Eric Regelin[1], Andreas Reiffs[1], Torsten Schmidt[1], Jan Schuladen[1], and Jos Lelieveld[1]

[1]Max Planck Institute for Chemistry, POB 3060, 55020 Mainz, Germany
[2]German Weather Service, Hohenpeißenberg, Germany
*now at Institute for Atmospheric Physics, Johannes-Gutenberg University, Mainz, Germany
**now at Dept. of Neurology, Johannes-Gutenberg University, Mainz, Germany
#now at Forschungszentrum Jülich, Germany

*Correspondence to*: Horst Fischer ([horst.fischer@mpic.de](horst.fischer@mpic.de))

**Abstract.** Hydrogen peroxide ($H_2O_2$) plays a significant role in the oxidizing capacity of the atmosphere. It is an efficient oxidant in the liquid phase, and serves as a temporary reservoir for the hydroxyl radical (OH), the most important oxidizing agent in the gas phase. Due to its high solubility, removal of $H_2O_2$ due to wet and dry deposition is efficient, being a sink of $HO_x$ (OH + $HO_2$) radicals. In the continental boundary layer, the $H_2O_2$ budget is controlled by photochemistry, transport and deposition processes. Here we use in-situ observations of $H_2O_2$, and account for chemical source and removal mechanisms to study the interplay between these processes. The data were obtained during five ground-based field campaigns across Europe from 2008 to 2014, and bring together observations in a boreal forest, two mountainous sites in Germany, and coastal sites in Spain and Cyprus. Most campaigns took place in the summer, while the measurements in the south-west of Spain took place in early winter. Diel variations in $H_2O_2$ are strongly site-dependent and indicate a significant altitude dependence. While boundary layer mixing ratios of $H_2O_2$ at low-level sites show classical diel cycles with lowest values in the early morning and maxima around local noon, diel profiles are reversed on mountainous sites due to transport from the nocturnal residual layer and the free troposphere. The concentration of hydrogen peroxide is largely governed by its main precursor, the hydroperoxy radical ($HO_2$), and shows significant anti-correlation with nitrogen oxides ($NO_x$) that remove $HO_2$. A budget calculation indicates that in all campaigns, the noontime photochemical production rate through the self-reaction of $HO_2$ radicals was much larger than photochemical loss due to reaction with OH and photolysis, and that dry deposition is the dominant loss mechanism. Estimated dry deposition velocities varied between approx. 1 and 6 cm/s, with relatively high values observed during the day in forested regions, indicating enhanced uptake of $H_2O_2$ by vegetation. In order to reproduce the change in $H_2O_2$ mixing ratios between sunrise and midday, a variable contribution from transport (10 – 100 %) is required to balance net photochemical production and deposition loss. Transport is most likely related to entrainment from the residual layer above the nocturnal boundary layer during the growth of the boundary layer in the morning.



## 1 Introduction

Hydrogen peroxide ($H_2O_2$) plays a pivotal role for the oxidizing capacity of the atmosphere. In hydrometeors and aqueous particles it oxidizes dissolved inorganic trace gases, while in the gas phase it serves as a reservoir species for the atmosphere's most important oxidizing agent, the hydroxyl radical (OH). Thus, $H_2O_2$ has a dual role as a secondary source for OH radicals

and an irreversible sink for $HO_x$ (OH + $HO_2$) due to its physical removal by wet and dry deposition. The atmospheric chemistry, concentration levels in the troposphere and measurement techniques used to observe $H_2O_2$, have been discussed in a number of review articles (Gunz and Hoffmann, 1990; Jackson and Hewitt, 1999; Lee et al., 2000; Vione et al., 2003; Reeves and Penkett, 2003). The dominant photochemical source of $H_2O_2$ is the recombination of two hydroperoxy radicals ($HO_2$):

$$HO_2 + HO_2 + M \rightarrow H_2O_2 + O_2 + M \tag{R1}$$

Here M represents a collision partner usually nitrogen ($N_2$), oxygen ($O_2$) or water vapour ($H_2O$). Note that the rate coefficient for R1 is a function of pressure p (due to its dependence on M) and the water vapour concentration [$H_2O$]. In general, the rate coefficient increases with p and $H_2O$ (Atkinson et al., 2004; http://iupac.pole-ether.fr). Additional production of $H_2O_2$ can result from the ozonolysis of alkenes (Sauer et al., 1999), in particular biogenic alkenes emitted from forests.

The formation of $H_2O_2$ according to R1 competes with the reaction of $HO_2$ with NO

$$HO_2 + NO \rightarrow OH + NO_2 \tag{R2}$$

one of the most important OH recycling reactions (Lelieveld et al., 2016) and an important step in photochemical ozone formation in the troposphere (Seinfeld and Pandis, 1997). Due to the competition for $HO_2$ described in R1 and R2 it is expected that $H_2O_2$ mixing ratios will show a dependence on ambient $NO_x$ (NO + $NO_2$) levels, with highest levels expected at lowest $NO_x$.

Photochemical loss of $H_2O_2$ is due to either reaction with OH (R3) or photolysis (R4), recycling $HO_x$ radicals:

$$H_2O_2 + OH \rightarrow HO_2 + H_2O \tag{R3}$$

$$H_2O_2 + h\nu \rightarrow 2\ OH \tag{R4}$$

Since reactions 3 and 4 recycle $HO_x$, they do not constitute an irreversible loss mechanism for $HO_x$ or $H_2O_2$, the latter being due to physical removal of $H_2O_2$ by wet and dry deposition. Due to its high Henry's law coefficient ($\sim 10^5$ molL$^{-1}$atm$^{-1}$), $H_2O_2$

is highly soluble in water and will be efficiently removed by rain and the deposition of fog (Klippel et al., 2011). Additionally, dry deposition rates with deposition velocities of the order of $1 - 5$ cms$^{-1}$ (see e.g. Table 6 in Stickler et al., 2007) lead to large losses of $H_2O_2$ in the boundary layer. Due to this strong surface sink, airborne observations often show increasing $H_2O_2$ mixing ratios with altitude, yielding a local maximum slightly above the boundary layer (Stickler et al., 2007; Klippel et al., 2011).

To understand the $H_2O_2$ budget and diurnal variability one has to consider all chemical and physical processes. Along with net

photochemical production (production minus loss) and deposition processes, horizontal and vertical transport have to be accounted for. In the absence of clouds, changes in the concentration of $H_2O_2$ can thus be described by equation 1:

$$\frac{d[H_2O_2]}{dt} = P_{chem} - L_{chem} + \frac{\omega_e\Delta[H_2O_2] - v_d[H_2O_2]}{BLH} - \nabla(v[H_2O_2]) \tag{Eq. 1}$$



Here $P_{chem}$ is the photochemical production of $H_2O_2$ by reaction 1, neglecting additional contributions from the ozonolysis of alkenes, while $L_{chem}$ is the loss due to reactions 3 and 4. The third term on the right-hand side of Eq. 1 describes vertical transport due to entrainment across the top of the boundary layer $\omega_e \Delta[H_2O_2]/BLH$ ($\omega_e$ is the entrainment velocity, $\Delta[H_2O_2]$ is the concentration difference between the boundary layer and the free troposphere, BLH is the boundary layer height) and dry deposition to the surface $v_d[H_2O_2]/BLH$ ($v_d$ is the deposition velocity). The final term in Eq. 1 describes the horizontal advection of $H_2O_2$ due to a horizontal gradient in $H_2O_2$ mixing ratios. The relative strength of the individual terms in Eq. 1 strongly depends on local conditions. In the free troposphere, dry deposition can be neglected and horizontal and vertical transport are small due to small concentration gradients on a regional scale. Thus, net photochemical tendencies ($P_{chem} - L_{chem}$) and precipitation largely determine the $H_2O_2$ concentrations in the free troposphere (Klippel et al., 2011). In the boundary layer, both transport and dry deposition play a significant role. Due to a rather invariant boundary layer height over the oceans and small horizontal $H_2O_2$ concentration gradients in the marine boundary layer, wet and dry deposition and net photochemical tendencies are the dominant processes affecting the $H_2O_2$ concentrations (Fischer et al., 2015). In the continental boundary layer, the situation can be complex, since all processes described in Eq. 1 are expected to play a role. Following up on previous studies in the free troposphere (Klippel et al., 2011) and the marine boundary layer (Fischer et al., 2015), we examine the influence of chemical and physical processes on the $H_2O_2$ budget at various locations in the continental boundary layer at various locations in Europe. We use in-situ observations of $H_2O_2$, its precursor ($HO_2$), sinks (R2 and R3), as well as measurements of species that are expected to influence $H_2O_2$ photochemistry (i.e. nitrogen oxides ($NO_x$) and ozone ($O_3$)), together with meteorological and boundary layer height information to study the $H_2O_2$ budgets. Overall, we use observations from five measurement campaigns spanning a latitude range from 61.5°N to 34.9°N between 2008 and 2014. With the exception of one campaign that was performed in early winter in southern Spain, all observations pertain to the summer. The main aim of this paper is to explore geographical differences in $H_2O_2$ mixing ratios and to what extent they are due to characteristics of the chemical environment, in particular with respect to $HO_x$ and $NO_x$ levels. Additionally, we investigate the role of transport and physical removal processes on $H_2O_2$ levels and diel variations. Rather than presenting individual timeseries, we will concentrate on diel variations, calculated from median values, being relatively less sensitive to individual events, e.g. precipitation or cloud processing. To illustrate atmospheric variability 25 – 75 % quartiles will be used. By using diel variations rather than time-series, we neglect the influence of variability in air mass origin and concentrate on the role of vertical transport due to boundary layer entrainment. Use of campaign averaged, diel profiles allows us to calculate median chemical tendencies and estimations of average deposition rates.

Section 2 describes the measurement sites, the techniques used for the in-situ measurements of $H_2O_2$, OH, $HO_2$, $NO_x$, $O_3$ and photolysis rates, and the derivation of the $H_2O_2$ photolysis rate that was not measured in all campaigns. In the results section (section 3) we discuss $H_2O_2$ mixing ratios, their relation to $NO_x$ and $HO_x$, diel variations and derive a $H_2O_2$ budget with respect to photochemical production and destruction, dry deposition and vertical entrainment. In the discussion (section 4), the limitations of our approach are discussed and results are compared to literature values.



## 2 Methods

### 2.1 Campaigns and observation sites

Between winter 2008 and summer 2014, we performed five measurement campaigns at various locations across Europe. In Table 1 a summary of the different campaigns, their location (latitude, longitude, height above sea level) and the time difference

between UTC and local noon is given. The location of the different campaigns is documented in Figure 1. The Diel Oxidant Mechanism in relation to Nitrogen Oxides (DOMINO) campaign was carried out at the El Arenosillo station (31.7°N, 6.7°W, 40 m asl) in the period between November 21 and December 8, 2008. El Arenosillo is located in the south-west of Spain approx. 200 m from the Atlantic Ocean. The site itself is situated in a national park. The city of Huelva, a large industrial complex, is situated 26 km to the NW and the Seville metropolitan area is 75 km to the NE. Details of the site and the

meteorological conditions including a characterisation of air mass origins can be found in Adame et al. (2014).

The Hyytiälä United Measurements of Photochemistry and Particles – Comprehensive Organic Particle and Environmental Chemistry (HUMPPA-COPEC) campaign was conducted at the boreal forest research station SMEAR II (Station for Measuring Ecosystem –Atmosphere Relation) in Hyytiälä, Finland (61.5°N, 24.17°E, 181 m asl) from July 12 to August 12, 2010. The site is situated in a large boreal forest, with the next major urban setting being Tampere approximately 50 km to the

SW of the site. Details of the site, the meteorology during HUMPPA and air mass origins can be found in Williams et al. (2011).

The PArticles and RAdicals Diel observations of the impact of urban and biogenic Emissions (PARADE) campaign was conducted at the Taunus Observatory on the Kleiner Feldberg mountain (50.22°N, 8.45°E, 825 m asl) in south-west Germany between August 15 and September 9, 2011. The site is close to the Rhine-Main area with the cities of Mainz 25 km to the

SSW, Wiesbaden 20 km to SW and Frankfurt 30 km to the SE. Details of the site, the meteorology during PARADE and air mass origins can be found in Li et al. (2015) and Sobanski et al. (2016).

The HOhenpeißenber Photochemistry Experiment (HOPE 2012) was conducted at the Global Atmospheric Watch (GAW) Metorological Observatory Hohenpeißenberg (47.48°N, 11°E, 988.8 m asl) in southern Germany between June 11 and July 13, 2012. This hilltop observatory operated by the German Weather Service is situated approximately 80 km SW of the

Bavarian capital Munich in a rural area. Details of the site and trace gas measurements from HOPE 2012 can be found in Novelli et al. (2017).

The CYprus PHotochemistry EXperiment (CYPHEX) was conducted on a hilltop site in north-western Cyprus at Ineia (34.96°N, 32.37°E, 650 m asl) during the period between July 7 and August 4, 2014. The site is situated in a rural area with no major population centres upwind in the W and NW directions. The distance to the Mediterranean Sea shoreline is

approximately 10 km.  Details about the site, the meteorology during CYPHEX and air mass origins can be found in Derstroff et al. (2017).



## 2.2 Trace gas measurements

During the campaigns discussed here $H_2O_2$ was measured with a commercial analyser (AL2001 CA, Aero Laser, Garmisch Partenkirchen, Germany) based on the wet chemical dual enzyme technique described by Lazarus et al. (1985, 1986). The instrument has been used previously for airborne (Stickler et al., 2007; Klippel et al., 2011) and ship-based (Fischer et al.,

2015) measurements and details of the instrument operation and performance are found in these publications. The detection limit of the instrument is of the order of 25 pptv at a time resolution (10 – 90%) of 30 s. The total uncertainty is typically of the order of 12 – 15 % (Fischer et al., 2015).

Nitrogen oxides (NO and $NO_2$) were measured with a highly sensitive two-channel chemiluminescence detector (CLD, ECO Physics CLD 790 SR, Duernten, Switzerland) during the DOMINO, HUMPPA, PARADE and CYPHEX campaigns. The

instrument has been previously used in a number of airborne and ship-based campaigns and is described in detail in Hosaynali Beygi et al. (2011). The time resolution is 1 s and typical detection limits are in the low pptv range with a total uncertainty of the order of 3 and 5 % for NO and $NO_2$, respectively. During HOPE $NO_x$ measurements were performed by the German Weather Service with a similar measurement technique.

Ozone was measured during all campaigns using a commercial UV Photometric $O_3$-Analyzer (model 49, Thermo Environment

Instruments, USA). The detection limit was typically 2 ppbv and the total uncertainty less than 5 %.

Measurements of OH and $HO_2$ radicals were conducted with the Max Planck Institute for Chemistry HORUS instrument based on laser induced fluorescence detection (Martinez et al., 2010; Hens et al., 2014). OH is detected directly, while $HO_2$ is measured indirectly as OH following conversion with NO via R2. Typical detection limits for OH and $HO_2$ are 9 x $10^5$ molec $cm^{-3}$ and 0.4 pptv, respectively. The total uncertainty is of the order of 30% (Hens et al., 2014). Note that since the HUMPPA

campaign in 2010 an inlet pre-injector (IPI) has been used to determine the OH background signal via a chemical modulation (Novelli et al., 2014). This technique was not used during the DOMINO campaign so that OH measurements from this campaign are considered to be an upper limit. Moreover, during PARADE, HOPE and CYPHEX $HO_2$ was measured using reduced amounts of NO sufficient to convert 10 – 30 % of the $HO_2$ but low enough to avoid conversion of $RO_2$ (Fuchs et al., 2011, Whalley et al., 2013). Previous measurements of $HO_2$ reported for DOMINO and HUMPPA are therefore an upper limit

as they are affected by a fractional measurement of $RO_2$. Crowley et al. (2018) suggest that on average 50 % of the daytime $HO_2$ signal during HUMPPA is due to a $RO_2$ interference caused by high NO additions. During later campaigns (PARADE, HOPE, CYPHEX) the reduction of NO addition resulted in significantly smaller $RO_2$ interferences of the order 12 to 15 % (Mallik et al., 2018).

During PARADE and CYPHEX photolysis rates for a large number of trace gases were measured with a commercial single

monochromator spectroradiometer (Metorologie Consult GmbH, Glashütten, Germany), while on all other campaigns $J(NO_2)$ was measured with a filter radiometer (Metorologie Consult GmbH, Glashütten, Germany). Based on a correlation analysis between measured $J(NO_2)$ and measured $J(H_2O_2)$ during PARADE and CYPHEX a second order correlation function was determined ($J(H_2O_2) = 0.015\ J(NO_2)^2 + 0.0004\ J(NO_2) + 6$ x $10^{-9}$, $R^2 = 0.99$), which was used to calculate $J(H_2O_2)$ from



measured J(NO$_2$) during DOMINO, HUMPPA and HOPE. We estimate the total uncertainty of the H$_2$O$_2$ photolysis rates obtained by this method to be of the order of 10%.

## 3 Results and discussions

### 3.1 Diel variations

Diel variations have been calculated for NOx, O$_3$, OH, HO$_2$, H$_2$O$_2$, and JNO$_2$ by binning the data into 30 min bins, calculating median values, 25 and 75 % quartiles and minimum and maximum values for each bin Fig. S1 – S5. We use the medians instead of means to be less sensitive to outlier values, e.g. due to measurements below the detection limit or rain events. For the same reason, we use quartiles instead of standard deviations. Table S1 in the supplement lists the data coverage (in %) for each species measured during the individual campaigns. Complete coverage (100%) refers to uninterrupted measurements

throughout the campaign time given in Table 1. In general, data coverage is less than 100% due to calibrations, instrument maintenance and failure. Figure 2 shows the H$_2$O$_2$ diel variations and solar zenith angle for DOMINO, HUMPPA, PARADE, HOPE and CYPHEX, respectively. Visual inspection of the H$_2$O$_2$ diel variations indicates two groups with different behaviour: sites on flat terrain like those encountered during DOMINO (Fig. 2a) and HUMPPA (Fig. 2b) versus hilltop sites probed during PARADE (Fig. 2c), HOPE (Fig. 2d) and CYPHEX (Fig. 2e). The first group (flat terrain) exhibits local minima in the early

morning hours between 5:30 – 8:30 UTC during DOMINO and between 4:30 – 7:30 UTC during HUMPPA, corresponding to local times between 6:00 and 9:00. Sunrise during DOMINO and HUMPPA was around 7:30 UTC and 4:00 UTC, respectively. These minima are followed by steep increases in the H$_2$O$_2$ mixing ratios reaching broad maxima between local noon and the early afternoon (DOMINO: 12:00 – 16:00 UTC; HUMPPA: 12:00 – 18:00 UTC), followed by a slow decrease during the late afternoon into the night. At these sites the daytime H$_2$O$_2$ mixing ratios are significantly higher than during the

night, and the diel variations are similar to those observed for O$_3$ (Fig. S1b and S2b). This is typical for the behaviour of a photochemically produced species in the continental boundary layer at a site with no significant orographic features. It is due to the interplay between net photochemical production during the day and strong deposition loss, scaling inversely with the variation of the boundary layer height.

The second group of sites is situated on hilltops and shows different characteristics. Although H$_2$O$_2$ mixing ratios during

PARADE (Fig. 2c), HOPE (Fig. 2d) and CYPHEX (Fig. 2e) exhibit similar local minima in the early morning hours and increasing mixing ratios afterwards with maximum values between noon and the early afternoon, the night-time mixing ratios are often higher than during the day. A similar evolution was observed for O$_3$ (Fig. S3b – S5b) and is typical for mountainous sites with up-slope air flow during the day due to local heating of the mountain slopes and descending air flow due to cooling during the night (Zaveri et al., 1995). Comparable H$_2$O$_2$ diel profiles have been described previously during observations at

Mauna Loa, Hawaii (Heikes, 1992) and at Izana, Tenerife (de Reus et al., 2005). The higher mixing ratios of H$_2$O$_2$ and O$_3$ during the night are generally due to sampling from higher altitudes (the nocturnal residual layer or the free troposphere), where mixing ratios for both species are expected to be higher as deposition losses are negligible.



## 3.2 Median values and dependence on $HO_x$ and $NO_x$

Median $H_2O_2$ mixing ratios averaged across the diel cycle vastly differ from site to site. Here we investigate the causes of these differences by plotting campaign median (25 – 75 quartiles) $H_2O_2$ mixing ratios versus median (and quartiles) $HO_2$ and $NO_x$,

respectively (Fig. 3 and 4). The lowest $H_2O_2$ mixing ratios are observed for DOMINO with median values (25 to 75 % quartiles given in parenthesis) of 58 pptv (37 – 91 pptv) for 24 h averages. Daytime mixing ratios (filtered by $J(NO_2) > 10^{-3}$ s$^{-1}$) are slightly higher: 72 pptv (49 – 94 pptv). This was expected since DOMINO is the only campaign that took place in the early winter, when $HO_x$ levels and thus the oxidizing capacity of the atmosphere are generally lower. Higher mixing ratios are obtained during HOPE 169 pptv (108 – 267 pptv), PARADE 270 pptv (148 – 585 pptv), HUMPPA 382 pptv (209 – 786 pptv)

and CYPHEX 601 pptv (420 – 936 pptv). Daytime only values during HUMPPA are higher than the 24 h averages (473 pptv (227 – 907 pptv)) similar to observations at DOMINO. This is in line with the discussion of the diel variations in section 3.1 were it was found that flat terrain sites exhibit higher $H_2O_2$ mixing ratios during the day compared to the night. For the mountainous sites, there is no significant difference between the 24 hour averages listed above and the daytime only values: HOPE 155 pptv (103 – 241 pptv), PARADE 230 pptv (153 – 452), CYPHEX 596 pptv (444 – 762 pptv).

These mixing ratios are consistent with previous observations over Europe, which indicated a general tendency for the highest mixing ratios in the summer season and lowest during winter; e.g. Morgan and Jackson (2002) observed a mean mixing ratio of 1.58 ppbv in June 1999 at Mace Head (Ireland) during the PARFORCE campaign and 0.23 ppbv in September 1998. This kind of seasonal variation is also observed at higher altitudes: Fels and Junkermann (1994) observed an average concentration of approx. 750 pptv of $H_2O_2$ in the summer of 1990 at an Alpine mountain station (Wank, Germany) while lower values of

185 ± 233 pptv were reported for February/March 2006 at the neighbouring Jungfraujoch (Switzerland) (Walker et al., 2006). Airborne observations in the continental boundary layer (below 2 km) over Europe confirm this tendency with mean (± 1σ-standarddeviation) mixing ratios of 0.55 ± 0.37 ppbv, 1.72 ± 1.34 ppbv, 1.74 ± 0.75 ppbv and 0.92 ± 0.47 ppbv during March 2004, July 2003, July 2007 and October 2006, respectively (Klippel et al., 2011).

With respect to diel variations, previous studies confirm the differences found here between mountainous sites and those at

flat terrain. Fischer et al. (1998) reported higher $H_2O_2$ mixing ratios at night (~ 2.4 ppbv) than during the day (2.1 ppbv) at the high-altitude site Izana (Tenerife) during July/August 1993. This result was confirmed in July/August 2002 at the same side by de Reus et al. (2005). Average daytime mixing ratios were 1.24 ppbv ± 0.38 ppbv increasing to 1.72 ± 0.55 ppbv during the night. Contrary to mountainous and flat continental sites, coastal sites often exhibit no or only weak diel variations (e.g. Sauer et al., 1997; Morgan and Jackson, 2002) in line with the observations during DOMINO. Strong diel variations with

daytime maxima have been reported for Tabua (Portugal) in June/July 1994 (night: < 15 pptv; day: 0.45 ± 0.33 ppbv) (Sauer et al., 2001), Zagreb (Croatia) in the summer of 2004 (night: 0.2 ± 0.35 ppbv; day: 0.4 ± 0.56 ppbv) (Acker et al., 2008), and at Waldstein (Germany) in July/August 2001 (night: ~ 0.1 ppbv; day: ~ 0.6 ppbv) (Ganzeveld et al., 2006; Valverde-Canossa





et al., 2006). In general, the $H_2O_2$ mixing ratios and diel variations reported in this study are in good agreement with previous observations for similar locations.

Based on airborne measurements in the continental boundary layer over Europe, Klippel et al. (2011) reported a significant latitudinal gradient of $H_2O_2$, with decreasing mixing ratios at increasing latitude, reflecting decreasing $HO_x$ and photochemical activity. This behaviour is only partly reproduced in the present study, indicating that other local effects have a strong influence on the mixing ratio of $H_2O_2$ at ground level. A suitable measure for the photochemical activity (or the oxidizing power of the lower troposphere) is the $HO_2$ concentration during the day, which is also a precursor of $H_2O_2$ according to R1. In Figure 3 we therefore plot the daytime median $H_2O_2$ and $HO_2$ mixing ratios against each other at the five measurement locations. The 25 and 75 % percentiles are also plotted. In general, the range of mixing ratios for an individual site is too small to yield significant correlations, but by comparing different environments, this limitation is removed. Figure 3 indicates a strong positive correlation between $H_2O_2$ mixing ratios and its precursor $HO_2$. Due to the quadratic dependency of the $H_2O_2$ production rate on $[HO_2]$ (R1) one expects that the mixing ratio of $H_2O_2$ exhibits a quadratic relation as well. The data in Fig. 3 can be subdivided into two groups at median $HO_2$ between 3 and 6 pptv (DOMINO, PARADE and HOPE) and those at higher $HO_2$ levels (18 – 24 pptv during CYPHEX and HUMPPA). Visual inspection suggests a roughly linear relation between $H_2O_2$ and $HO_2$. This is confirmed by a linear regression analysis based on median values, which yields a regression coefficient $R^2$ of 0.73. Plotting $H_2O_2$ versus $(HO_2)^2$ (not shown) yields a smaller $R^2$ of 0.61. This lower $R^2$ is largely due to the HUMPPA data point whereby a lower $H_2O_2$ mixing ratio was measured at higher $HO_2$ compared to CYPHEX. As reported by Hens et al. (2014) $HO_2$ measurements with the HORUS instrument are prone to interferences from peroxy radicals in particular of alkene-based peroxy radicals, which are expected to be most abundant in forest environments, e.g. during HUMPPA. In a recent modelling study, Crowley et al. (2018) determined the contribution of $RO_2$ to the measured $HO_2$ during the daylight hours to be of the order of 50 %. If we correct the HUMPPA data for this potential interference, median daytime $HO_2$ is reduced from 24 pptv (14 to 35 pptv for the 25 – 75 % percentiles) to 12 pptv (7 – 17.5 pptv). Thus, the data point for HUMPPA in Fig. 3 shifts to the left of the CYPHEX data point. A regression analysis of $H_2O_2$ versus $(HO_2)^2$ (not shown) with the reduced HUMPPA $HO_2$ results in $R^2$ of 0.9, yielding a much better agreement with the hypothesis that the measured $H_2O_2$ follows a quadratic dependence on $HO_2$. For further calculations, we have used the corrected HUMPPA $HO_2$ data. For DOMINO the $HO_2$ observations that were also measured at high NO additions were not corrected due to very low concentrations of biogenic VOCs (Sinha et al., 2012).

Since the concentration of $H_2O_2$ according to Eq. 4 depends strongly on $HO_2$ it is to be expected that the competing reaction of $HO_2$ with NO (R3) will have also have an effect. In Figure 4, we therefore plot the median mixing ratio (25 – 75 percentiles) of $H_2O_2$ versus median mixing ratio (25 – 75 percentiles) of $NO_x$ at the five measurement locations. Please note that contrary to Fig. 3 we use data obtained during both day and night. Restriction of the analysis to daytime data only, as has been done in Fig. 3, will not change the results. As can be expected there is a negative correlation between $H_2O_2$ and $NO_x$, with the highest $H_2O_2$ mixing ratio observed at the lowest $NO_x$ values.



Besides being dependent on $HO_2$ and $NO_x$, $H_2O_2$ mixing ratios also show a positive (linear) correlation with $O_3$ (Fig. S6), which is an indication for the dependence of $H_2O_2$ on photochemical activity. It is expected that higher $O_3$ is accompanied by higher $HO_x$ levels and thus an increasing $H_2O_2$ production rate.

Overall, it can be concluded that the $H_2O_2$ mixing ratios strongly depend on local oxidation rates represented by $HO_x$ and $O_3$ levels. Higher photochemical activity leads to higher concentrations of $H_2O_2$. Nitrogen oxides play a key role in modulating the $HO_x$ partitioning and thus affect $H_2O_2$ levels indirectly by influencing the $HO_2$ concentrations available. In order to study the role of physical processes (deposition and transport) on local $H_2O_2$ mixing ratio levels, we will next evaluate the $H_2O_2$ budget according to Eq. 1.

## 3.3 Hydrogen peroxide budgets

Equation 1 describes the temporal evolution of $H_2O_2$ neglecting scavenging by particles, cloud processing and wet deposition. Rain events and cloud processing that could result in total $H_2O_2$ removal were rare during the campaigns. Median values and 25 and 75 % percentiles do not include such events. Therefore, we can neglect wet deposition in the analysis of Eq. 1. In the following, we concentrate on the observed increases of $H_2O_2$ during the period between sunrise and midday. During this period, net photochemical production, dry deposition and vertical entrainment associated with the growth of the boundary layer are expected to influence the observed change in $H_2O_2$. Based on a comparison of the mean observed change ($dH_2O_2/dt$ in pptv s$^{-1}$) with calculations of the mean net production rate ($P_{chem} - L_{chem}$) and the deposition loss, we estimate potential entrainment rates during the growth of the boundary layer, neglecting horizontal advection.

The chemical production (Eq. 2) and destruction (Eq. 3) rates for $H_2O_2$ in pptv/s are shown as a function of local time in Fig. 5a and b, respectively:

$$P_{chem} = k(R1)[HO_2]^2 \tag{Eq. 2}$$

$$L_{chem} = (k(R3)[OH] + J(H_2O_2))[H_2O_2] \tag{Eq. 3}$$

The calculation of the production term according to Eq. 2 is based on the rate coefficient $k(R1)$ following the IUPAC recommendation, which takes into account an enhancement of the rate coefficient by water vapour (Atkinson et al., 2004; http://iupac.pole-ether.fr). Measured water vapour concentrations varied between 0.9% (DOMINO) and 2.2% (HUMPPA) leading to enhancement factors of the reaction coefficient of R1 between 1.5 and 2.2. However, it is the difference in median $HO_2$ concentrations that leads to the large variability in $P_{chem}$ derived for the different campaigns (Fig. 5a). Maximum noontime $H_2O_2$ production rates were 0.0015 pptv/s (DOMINO), 0.004 pptv/s (PARADE), 0.017 pptv/s (HOPE), 0.04 pptv/s (HUMPPA) and 0.13 pptv/s (CYPHEX). This variation by two orders of magnitude reflects the dependence of $P_{chem}$ on the $HO_2$ precursor concentrations (Fig. 3), which are highest for those sites with the lowest $NO_x$ concentrations (Fig. 4). Median nighttime production was ~ 0 pptv/s for DOMINO, PARADE, HOPE and CYPHEX, but between 0.01 and 0.02 pptv/s during HUMPPA, due to elevated $HO_2$ concentrations during the night (~ 20 pptv in Fig. S2d), which is most likely an artefact due to a $RO_2$ interference of the $HO_2$ measurements (Crowley et al. 2018). The total uncertainty is calculated by error propagation:





$$\Delta y = \sqrt{\Sigma [(\frac{\partial y}{\partial x})^2 * \Delta x^2]} \qquad \text{(Eq. 4)}$$

The largest contribution to the overall uncertainty is from atmospheric variability calculated from the 25 – 75 % percentiles of the input parameters $H_2O_2$ and $HO_2$ used to calculate $P_{chem}$ according to Eq. 2. The uncertainty in the rate constant is neglected, since it is much smaller that the atmospheric variability of the precursors. The uncertainty of $P_{chem}$ is ± 61 %, ± 80 %, ± 90 %,

± 80 %, and ± 35 % for PARADE, DOMINO, HUMPPA, HOPE and CYPHEX, respectively.

Absolute differences in the photochemical destruction rates (Eq. 3) for the individual campaigns differ by a factor of 10. Maximum $L_{chem}$ values for DOMINO, PARADE, HOPE, HUMPPA and CYPHEX are -0.001 pptv/s, -0.001 pptv/s, -0.0025 pptv/s, -0.0046 pptv/s and -0.015 pptv/s, respectively. During the night, photochemical loss was zero during all campaigns. Photolysis (R4) is the dominant photochemical $H_2O_2$ sink during CYPHEX (~ 70 %), HUMPPA (~ 75 %) and PARADE (~

90 %). During DOMINO and HOPE, photolysis and reaction with OH (R3) are of the same order of magnitude. Calculation of the uncertainty of $L_{chem}$ according to Eq. 4 is based on the atmospheric variability of the variables in Eq. 3 (i.e. $H_2O_2$, OH and $J(H_2O_2)$) yields ± 86 %, ± 62 %, ± 76 %, ± 72 %, and ± 39 % for PARADE, DOMINO, HUMPPA, HOPE and CYPHEX, respectively.

Due to the much higher value of $P_{chem}$ relative to $L_{chem}$ (at least one order of magnitude) the net chemical production rate (NPR

= $P_{chem}$ - $L_{chem}$) is similar to Fig. 5a. The only exception is DOMINO, were the photochemical sources and sinks of $H_2O_2$ are almost balanced. It is interesting to evaluate observed $H_2O_2$ trends according to Eq. 1.

In order to calculate the effect of dry deposition we use two different approaches. First, we estimate the dry deposition loss rate constant from the decrease of $H_2O_2$ mixing ratios during the night, when photochemical production and loss, as well as vertical entrainment can be assumed to be negligible. This estimate of the dry deposition sink might be a lower limit, since it

will neglect thermal turbulence and dry deposition due to stomatal up-take by vegetation, which occurs only during the day and does not take into account day-night changes in the rate of turbulent transport to the ground (e.g. Nguyen et al., 2015). In order to account for the contribution of stomatal uptake, we will also estimate dry deposition loss during local noontime. During this time of the day, which is generally close to the maximum $H_2O_2$ mixing ratios, it can be assumed that the daytime boundary layer is fully established and vertical intrusion is at a minimum. Concentrating on periods with $d[H_2O_2]/dt \sim 0$, only

net chemical production (NPR), dry deposition and horizontal advection will influence the concentration of $H_2O_2$.

For these two cases we calculate an average loss rate $k_d$ constant according to

$$k_d = \frac{\frac{dH_2O_2}{dt}}{H_2O_2} \ [\text{s}^{-1}] \qquad \text{(Eq.5)}$$

and the deposition velocity $v_d$ as

$$v_d = \frac{k_d * BLH}{x} \ [\text{cm s}^{-1}] \qquad \text{(Eq.6)}$$

with BLH the boundary layer height in cm. The factor $x$ takes into account a potential gradient in the mixing ratio profile. During the night $x = 2$, assuming a linear increase of the mixing ratio with height in the nocturnal boundary layer (Shepson et al., 1992). During the day, we assume that the boundary layer is well mixed and $x$ is equal to 1.



Table 2 lists the time span over which $d[H_2O_2]/dt$ was analysed, the change in $H_2O_2$ mixing ratio $\Delta H_2O_2$, the mean $d[H_2O_2]/dt$, $k_d$, BLH and $v_d$. Values for the BLH during the night were taken from van Stratum et al. (2012) for DOMINO, Ouwersloot et al. (2012) for HUMPPA, and Berkes et al. (2016) for PARADE. For HOPE boundary layer height measurements are not available, so the nocturnal BLH was estimated to be 200 m, similar to measurements at the other sites. We assign an uncertainty

of 20 % to all BLH values. Note that for CYPHEX this method cannot be applied, since the nighttime mixing ratios of $H_2O_2$ exhibit a tendency to increase while the hilltop extends into the free troposphere. Here, the observed decrease of $H_2O_2$ in the early morning occurs during sunlit hours. For the estimation of the night-time deposition velocities we follow the approach of Shepson et al. (1992) and Hall and Claiborn (1997) assuming that the deposition loss is a first-order loss process resulting in an exponential decrease of $\ln(H_2O_2)$ (Hall and Claiborn, 1997). Additionally, we assume a linear $H_2O_2$ gradient throughout the

nocturnal boundary layer (Shepson et al., 1992). Please note that nighttime production of $H_2O_2$ due to the ozonolysis of alkenes is neglected in this approach, leading to a potential underestimation of the deposition velocities in particular in environments with large biogenic emissions as during HUMPPA. Estimated deposition velocities varied between 0.18 and 0.6 cm/s (Table 2) with a total uncertainty between $\pm$ 53 and $\pm$ 105 %. These values are similar to values for the $H_2O_2$ dry deposition velocity found in the literature. Walcek (1987) reported a value of 1 cms$^{-1}$ over the north-east of the USA while Baer and Nester (1992)

estimated an average $v_d$ of 1.5 cm s$^{-1}$ for the upper Rhine Valley (Germany). From airborne measurements over the tropical rainforest in Suriname Stickler et al. (2007) deduced a $H_2O_2$ deposition velocity of 1.35 cm s$^{-1}$. Higher values of $v_d$ up to 5 - 10 cm s$^{-1}$ are reported over forested regions due to enhanced up-take by stomatal openings (Hall and Claiborn, 1998; Valverde-Canossa et al., 2006; Nguyen et al., 2015). The nighttime $v_d$ values listed in Table 2 can thus be assumed to be lower limits of daytime values, since the effect of vegetation and enhanced turbulence is not accounted for.

The daytime analysis of $v_d$ has been performed for periods of the day in which $H_2O_2$ can be assumed to be in photostationary state ($dH_2O_2/dt = 0$). This criterion is generally met between local noon and the early afternoon when the BLH is highest and vertical entrainment can be neglected. For the calculation of $v_d$ in Table 3 we assume that net $H_2O_2$ production (NPR = $P_{chem}$ $-$ $L_{chem}$) is balanced by dry deposition loss. The deposition velocities for DOMINO (0.56 cm s$^{-1}$; uncertainty $\pm$ 85 %) and PARADE (0.6 cm/s; uncertainty $\pm$ 98 %) are about a factor of 2 to 3 higher than the night-time values for these sites

documented in Table 2, while significantly higher daytime $v_d$ values (factor of 10 to 20) (Table 3) are calculated for HUMPPA and HOPE. The value of $v_d$ (day) for CYPHEX of 2.1 cm s$^{-1}$ (uncertainty $\pm$ 50 %) is within the range of observation at other sites both investigated here and those cited in the literature. The daytime $v_d$ value obtained for HOPE (6 cm/s; uncertainty $\pm$ 93%) is also within the range of values reported in the literature for forested environments (Hall and Claiborn, 1998; Valverde-Canossa et al., 2006; Nguyen et al., 2015), while the value obtained for HUMPPA (6.04 cm s$^{-1}$; uncertainty $\pm$ 100%) is

comparable to the high values reported for a boreal forest in Canada (Hall and Claiborn, 1997). Note that uncertainties were calculated according to Eq. 4, taken into account the variability of all input variables to Eq. 1 derived from the 25 – 75 % range and an uncertainty of 20 % for the boundary layer height.

To evaluate Eq. 1 from sunrise to mid-day during the five campaigns we use the net photochemical production of $H_2O_2$ presented in Figure 5 and calculate the deposition loss during the increase of the boundary layer height. For this calculation,





we linearly interpolate the deposition velocity between the nighttime values presented in Table 2 and the noontime values presented in Table 3. For CYPHEX we use an average of all nighttime deposition velocities presented in Table 3. As mentioned before, during this period it is expected that vertical entrainment due to an increasing boundary layer height and horizontal advection will also have an effect on the temporal evolution of $H_2O_2$. The mean rate of $d[H_2O_2]/dt$ is derived from the observed

$H_2O_2$ mixing ratio increase from the early morning minimum up to the maximum around local noon in Fig. 2. Average net photochemical production rates $(P_{chem} - L_{chem})$ and dry deposition loss rates over the periods for which $d[H_2O_2]/dt$ was analysed were derived from Eq. 2 and 3 and Eq. 7, respectively.

$$L_{dep} = \frac{v_d\,[H_2O_2]}{BLH} \qquad\qquad\qquad\qquad \text{(Eq. 7)}$$

The residual $(d[H_2O_2]/dt - ((P_{chem} - L_{chem}) - L_{dep}))$ according to Eq. 1 is a measure for gain or loss of $H_2O_2$ due to the combination

of vertical entrainment and horizontal advection. Table 4 lists the time periods over which $d[H_2O_2]/dt$ was analysed, the mean $H_2O_2$ mixing ratio [pptv], mean $d[H_2O_2]/dt$ [pptv/h], mean net photochemical production rate $(P_{chem} - L_{chem})$ [pptv/h], the mean boundary layer height (BLH) [cm], the deposition loss rate $(L_{dep})$ [pptv/h] and the transport rate $(P_{trans})$ [pptv/h]. Uncertainties in percentage were calculated from Eq. 4 based on the variabilities of the input variables. Positive residuals indicate vertical entrainment or advection of higher $H_2O_2$ mixing ratios, negative values indicate dilution. The budget of net photochemical

production, deposition loss, observed change in $H_2O_2$ mixing ratios from sunrise to noon and the inferred residual transport are graphically shown in Fig. 6. For DOMINO the calculated net photochemical production (1.3 pptv/h) is of the same order of magnitude as the loss rate due to deposition $(L_{dep} = -0.96$ pptv/h), indicating a balance between sources and sinks of $H_2O_2$. Thus the observed increase of $H_2O_2$ (7.9 pptv/h) during the morning is almost completely due to transport (7.6 pptv/h).

During PARADE the net production is 5.4 pptv/h, which is also largely balanced by deposition loss (-4.4 pptv/h), resulting in

a positive residual indicating a missing source of the order of 10.5 pptv/h. Since the PARADE site is on a hilltop it is likely that entrainment from the residual layer is responsible for this transport.

During HUMPPA the net photochemical production of 53.16 pptv $h^{-1}$ is only slightly smaller than the deposition loss (- 56.9 pptv/h) resulting in a rather large entrainment rate of the order of 114.2 pptv $h^{-1}$ required to explain the observed $H_2O_2$ increase. Since the surrounding area is rather homogeneous (Williams et al., 2011), we assume that this transport is due to vertical

entrainment from the residual layer during the rise of the boundary layer height. The deduced entrainment rate of 118.3 pptv $h^{-1}$ is an upper limit since we most likely underestimate the net production rate of $H_2O_2$. Axinte (2016) estimated that the ozonolysis of terpenes in the boreal forest would lead to an additional $H_2O_2$ production of the order of 8.3 pptv $^{s-1}$ enhancing the net production by 7.5%.

Slightly higher net production compared to deposition losses are observed for HOPE (net production = 27.3 pptv $h^{-1}$, deposition

loss = -20.4 pptv $h^{-1}$). This yields a contribution of 12.9 pptv $h^{-1}$ from transport. Since HOPE was performed on a mountaintop, we assume that this increase is due to vertical entrainment during the growth of the boundary layer. Contrary to the other sites discussed above where close to 100 % of the mourning increase in $H_2O_2$ was due to transport, this contribution is only 65 % during HOPE.



During CHYPHEX, the net production of 259 pptv h$^{-1}$ is only partly balanced by a dry deposition loss of -200.7 pptv h$^{-1}$. Thus, the photochemical production of $H_2O_2$ minus deposition (58.3 pptv h$^{-1}$) is slightly larger than the observed increase during the early morning (31.7 pptv h$^{-1}$) yielding a negative residual of -26.5 pptv h$^{-1}$, indicating a dilution. Since CYPHEX was performed at a high altitude coastal cite affected by a land sea breeze, it is likely that the advection of marine air masses with slightly lower $H_2O_2$ mixing ratios is responsible for this dilution effect.

## 4. Discussion

Besides the large uncertainty resulting from atmospheric variability affecting the median profiles (see Table 2 to 4), this analysis is also influenced by uncertainties associated with respect to data coverage and limitations in the method to derive deposition velocities and subsequently transport rates in particular during the day. These limitations will be discussed in the context of the HUMPPA campaign. This campaign is particularly suitable for this purpose since two other studies have been published that specifically address the temporal evolution of $H_2O_2$ using a box model (Crowley et al., 2018) and the contribution of entrainment to the early morning increase of $O_3$ (Ouwersloot et al., 2012) for this particular campaign. Results from HUMPPA will also apply to the other campaigns discussed here.

Uncertainties due to missing data are mainly due to temporally incomplete measurements of radical species. Table S1 indicates that during HUMPPA data coverage for $HO_2$ was only 14.7 %, so that it is questionable if a median diel cycle based on this relatively small dataset is representative for the whole campaign. Additionally, a potential interference in the $HO_2$ observations by $RO_2$ radicals (Hens et al., 2014) will also affect the $H_2O_2$ production rate. Crowley et al. (2018) used a box model to study PAA, PAN and $H_2O_2$ during HUMPPA, deriving $HO_2$ concentrations that fit the temporal evolution of these species between July 20 and August 12, 2010. Modelled diel averaged $HO_2$ profiles indicate maximum noontime values of 20 pptv roughly 50 % lower than mean observation (35 pptv), which is due to a modelled $RO_2$ interference (see Fig. 9 of Crowley et al., 2018). In our study, we thus assumed a 50 % interference on the $HO_2$ observations, yielding median $HO_2$ mixing ratios of 18 pptv at noon, 10 % lower than the average of Crowley et al. (2018). This indicates that the limited measurements of $HO_2$ during HUMPPA that cover approximately 12 days are in good agreement with the modelled data, that cover a longer period (23 days). According to Eq. 2 using the modelled $HO_2$ data from Crowley et al. (2018) leads to an approximately 23 % higher $H_2O_2$ production rate. The main effect of this higher production rate would be a higher deposition velocity derived from the steady state assumption around noon, yielding a deposition velocity of 7.4 cm/s instead of 6 cm/s, similar to the maximum value of 8.4 cm/s used by Crowley et al. (2018). The deposition velocity derived during the night by Crowley et al. (2018) is slightly larger than our estimate (0.8 cm/s vs. 0.6 cm/s), since Crowley et al. took the night-time production of $H_2O_2$ via ozonolysis of terpenes into account, which was not considered for in this study, and which leads to an underestimation of the nocturnal deposition loss in this study. Since in our study the inferred entrainment rate strongly depends on the deposition sink, uncertainties in derived deposition velocities will linearly affect the entrainment flux needed to explain the morning rise in $H_2O_2$. Note that the deposition velocities used by us and Crowley et al. (2018) are in rather good agreement with observation-



based estimates published in the literature (Hall and Claiburn, 1998; Valverde-Canossa et al., 2006, Nguyen et al., 2015), but are much higher than values used in the EMAC model for the boreal forest in Finland (~0.2 cm/s at night and 0.8 – 1 cm/s during the day) (Jöckel et al., 2016). Using these low deposition velocities would yield a deposition loss of only 8.9 pptv/h instead of 118 pptv/h and thus a transport contribution of only 8.7 pptv/h (7.8 % of the mourning increase). This illustrates that the $H_2O_2$ budget terms for deposition and transport in this study are highly coupled and depend strongly on the deposition velocities used.

Another potential error source are trends in $H_2O_2$ mixing ratios over the campaign. While this study covers the whole period of the HUMPPA campaign (July 12 until August 12, 2010) the model study by Crowley et al. (2018) started on July 20, 2010, missing the first week of the campaign. During this warm period, noontime $H_2O_2$ mixing ratios were higher than during the rest of the campaign, affecting the median increase after sunrise. Therefore, we obtained larger values of $d[H_2O_2]/dt$ over the whole campaign, compared to the reduced period used by Crowley et al. (2018). Note that a smaller value for $d[H_2O_2]/dt$ at constant net-production and deposition loss yields a smaller residual, i.e. less transport. This is the reason, that Crowley et al. (2018) did not include transport in their study.

In general, the results from our study are in good agreement with a $H_2O_2$ budget calculation for a coniferous forest in southern Germany based on a single-column chemistry-climate model made by Ganzeveld et al. (2006). They conclude that turbulent exchange is similar in magnitude to the deposition loss, and much larger than net photochemical production. Since $H_2O_2$ and $O_3$ have similar vertical profiles between the surface and the top of the boundary layer due to strong depositional sinks at the surface, they should behave similar with respect to entrainment. Ouwersloot et al. (2012) simulated the $O_3$ budget during HUMPPA with a single column model, taking into account photochemical production, depositional loss and vertical transport, indicating that the rise in boundary layer height in the early morning and the subsequent in-mixing of residual layer air is responsible for the majority of the observed $O_3$ increase.

The potential role of entrainment can also be illustrated by a simple scheme taking into account a two-box mixing process. If we assume that the $H_2O_2$ mixing ratio in the residual layer during the night is uniform and constant due to the absence of sinks (no photochemical production or loss and no deposition due to its isolation from the surface by the nocturnal inversion), this air will be mixed with air masses in the nocturnal boundary layer during the early morning rise in the BLH. The $H_2O_2$ mixing ratio in the nocturnal boundary layer just before sunrise is 260 pptv during HUMPPA. We further assume that the $H_2O_2$ mixing ratio in the residual layer is a remnant from the previous day with a mixing ratio of 600 pptv measured in the late afternoon at 16:15 UTC (17:45 local). For simplicity we assume that the height of the nocturnal boundary layer is 200 m and the height of the residual layer corresponds to the top of boundary layer (1500 m), yielding a depth of the residual layer of 1300 m. Mixing of these two boxes during the morning over a period of 6 hours and taking into account the depth of both layers yields a $H_2O_2$ increase of 49 pptv/h, which is a factor of 2 smaller than the value of 114 pptv/h derived in Table 4. This difference might be due to the restriction to two boxes, neglecting additional entrainment from the free troposphere.



## 5. Conclusions

The budget of hydrogen peroxide in the continental boundary layer is defined by the balance between photochemical production and loss, physical removal by dry and wet deposition, as well as vertical entrainment into the boundary layer and horizontal advection. We used measurements of $H_2O_2$, its precursor $HO_2$ and sinks (OH, $J(H_2O_2)$) at five European sites to
calculate net photochemical production. Assuming horizontal homogeneity and negligible rainout, we estimated both the dry deposition loss and the entrainment rate. In general, absolute mixing ratios of $H_2O_2$ exhibit an inverse relation to local NOx levels. The net production is a strong function of $HO_2$, and thus extremely sensitive to interferences in the measurement of this radical (Hens et al., 2014; Crowley et al., 2018). Calculated photochemical production rates generally exceed photochemical loss rates by at least an order of magnitude at all sites, except for one observation during the winter season (DOMINO) where
production and loss are approximately equivalent. Estimates of deposition velocities during the night are of the order of 0.16 – 0.6 cm/s and thus at the lower end of values reported in the literature (Walcek 1987; Baer and Nester, 1992; Stickler et al., 2007; Hall and Claiborn, 1998; Valverde-Canossa et al., 2006; Nguyen et al., 2015). This is to be expected since deposition of $H_2O_2$ during the day is often enhanced by stomatal uptake (Hall and Claiborn, 1998; Valverde-Canossa et al., 2006; Nguyen et al., 2015). Daytime deposition rates during the five campaigns are consistently higher in forested areas and reach values of
~ 6 cm/s, in good agreement with literature values (Hall and Claiborn, 1998; Valverde-Canossa et al., 2006; Nguyen et al., 2015). Using the individual terms for $H_2O_2$ photochemical production, photochemical loss and dry deposition we could show that the early morning rise in $H_2O_2$ mixing ratios is significantly influenced by dynamical processes. For DOMINO, HUMPPA and PARADE transport is responsible for almost 100 % of the observed early morning increase in $H_2O_2$. Smaller contributions of transport are derived HOPE (65 %) and CYPHEX (10 %). This transport is most likely related to vertical entrainment from
the residual layer during the early morning rise of the boundary layer height. As shown by aircraft measurements, strong deposition at the surface leads to increasing $H_2O_2$ mixing ratios with altitude up to the top of the boundary layer (Klippel et al., 2011), so that the entrainment during the early morning is a source of $H_2O_2$ (Fischer et al., 2015).

The findings of this study are in general agreement with previous studies of trace gas budgets for $H_2O_2$ (Ganzeveld et al., 2006; Stickler et al., 2007) and $O_3$ (Ouwersloot et al., 2012; Kaser et al., 2017) in the continental boundary layer that emphasize the
significant contribution of vertical entrainment in particular during the early morning hours. Nevertheless, the findings are rather qualitative since quantitative results strongly depend on the deposition velocity used in the budget calculations. In principle, the photochemical production and loss of $H_2O_2$ can be quantified by accurate local in-situ measurements of precursors (mainly $HO_2$) and losses due to photolysis and reaction with measured OH. The balance of net photochemical production, dry deposition and transport strongly depends on an accurate determination of the deposition velocity and its
temporal evolution. Point measurements, as presented here, suffer from strong limitations in deriving deposition velocities and subsequently potential transport contributions to local budgets. Future studies should therefore include vertically resolved measurements, preferentially from the surface to the top of the boundary layer, and/or include flux measurements of the species of interest.




**Author contributions.** HF, JNC, AP and JL designed the study. RA, HB, JNC, CE, SG, SH, HH, KH, RK, DK, CM, MM, AN, UW, CPD, ER, AR, TS, and JS conducted and processed the measurements. HF prepared the manuscript with contributions from all co-authors.

**Data availability.** Readers who are interested in the data should contact Horst Fischer (horst.fischer@mpic.de).





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



Table 1: Observational sites

| Campaign | Location | Duration | Latitude | Longitude | Altitude | Local time |
|---|---|---|---|---|---|---|
| DOMINO | El Arenosillo, Spain | Nov 21- Dec 8, 2008 | 31.7°N | 6.7°W | 40 m asl | UTC - 26 m |
| HUMPPA | Hyytiälä, Finland | Jul 12 – Aug 12, 2010 | 61.5°N | 24.1°E | 181 m asl | UTC + 96 m |
| PARADE | Kleiner Feldberg, Germany | Aug 15 – Sep 8, 2011 | 50.2°N | 8.4°E | 825 m asl | UTC + 33 m |
| HOPE | Hohenpeißenberg, Germany | June 11 – Jul 13, 2012 | 47.5°N | 11°E | 988 m asl | UTC + 44 m |
| CYPHEX | Ineia, Cyprus | Jul 7 – Aug 4, 2014 | 34.9°N | 32.4°E | 650 m asl | UTC + 128 m |

5    Table 2: Calculation of nighttime dry deposition loss rate $k_d$ and the deposition velocity $v_d$. Uncertainties are reported in %.

| Campaign | Time span | $\Delta H_2O_2$ [pptv] | $dH_2O_2/dt$ [pptv/s] | $k_d$ [1/s] | BLH [cm] | $v_d$ [cm/s] |
|---|---|---|---|---|---|---|
| DOMINO | 0:45 – 4:45 | -12 (±46) | 0.00008 (±46) | 0.000017 (±64) | 20000 (±20) | 0.16 (±67) |
| HUMPPA | 00:45 - 4:45 | -226 (±60) | 0.0156 (±60) | 0.000059 (±84) | 20000 (±20) | 0.6 (±86) |
| PARADE | 1:15 – 5:15 | -208 (±74) | 0.014 (±74) | 0.000032 (±103) | 17500 (±20) | 0.3 (±105) |
| HOPE | 2:45 – 4:45 | -69 (±35) | 0.0095 (±35) | 0.000058 (±49) | 20000 (±20) | 0.6 (±53) |

Table 3: Calculation of daytime dry deposition loss rate $k_d$ and the deposition velocity $v_d$. Uncertainties are reported in %.

| Campaign | Time span | Mean $H_2O_2$ [pptv] | NPR [pptv/s] | $k_d$ [1/s] | BLH [cm] | $v_d$ [cm/s] |
|---|---|---|---|---|---|---|
| DOMINO | 12:45 – 14:45 | 80 (±46) | 0.0003 (±83) | 0.000004 (±64) | 140000 (±20) | 0.56 (±85) |
| HUMPPA | 13:15 – 15:15 | 745 (±60) | 0.03 (±98) | 0.000084 (±84) | 150000 (±20) | 6.04 (±100) |
| PARADE | 14:45 – 16:45 | 258 (±74) | 0.0013 (±96) | 0.000005 (±103) | 130000 (±20) | 0.6 (±98) |
| HOPE | 14:15 – 16:15 | 222 (±35) | 0.009 (±91) | 0.00004 (±49) | 150000 (±20) | 6 (±93) |
| CYPHEX | 11:45 – 13:45 | 664 (±28) | 0.055 (±46) | 0.00008 (±39) | 25000 (±20) | 2.1 (±50) |

Table 4: Calculation of dry deposition loss $L_{dep}$ and entrainment rate $P_{ent}$. Uncertainties are reported in %.

| Campaign | Time span | $dH_2O_2/dt$ [pptv/h] | Mean BLH [cm] | NPR [pptv/h] | $L_{dep}$ [pptv/h] | $P_{trans}$ [pptv/h] |
|---|---|---|---|---|---|---|
| DOMINO | 7:45 – 13:15 | 7.9 (±46) | 80000 (±20) | 1.3 (±83) | -0.96 (±75) | 7.6 (±120) |
| HUMPPA | 7:15 – 13:15 | 110.5 (±60) | 85000 (±20) | 53.2 (±98) | -56.9 (±118) | 114.2 (±164) |
| PARADE | 10:15 – 14:45 | 11.5 (±74) | 73750 (±20) | 5.4 (±96) | -4.4 (±124) | 10.5 (±173) |
| HOPE | 8:15 – 13:15 | 19.8 (±35) | 85000 (±20) | 27.3 (±91) | -20.4 (±101) | 12.9 (±140) |
| CYPHEX | 5:45 – 13:15 | 31.7 (±28) | 12500 (±20) | 259 (±46) | -200.7 (±60) | -26.5 (±80) |




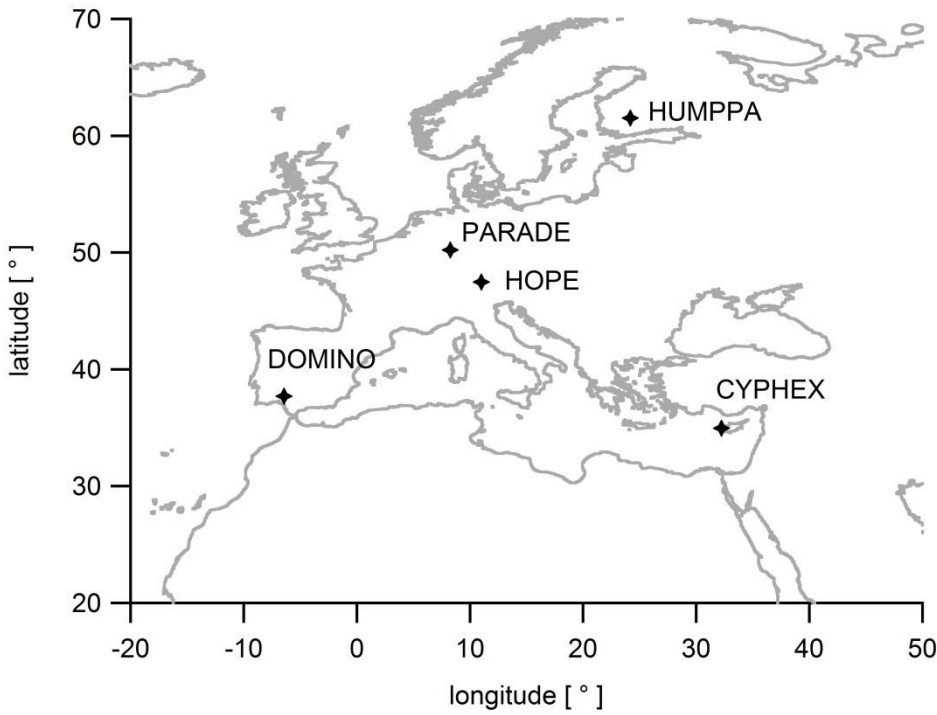

Figure 1: Locations of the different campaigns performed between 2008 and 2014 in Europe.

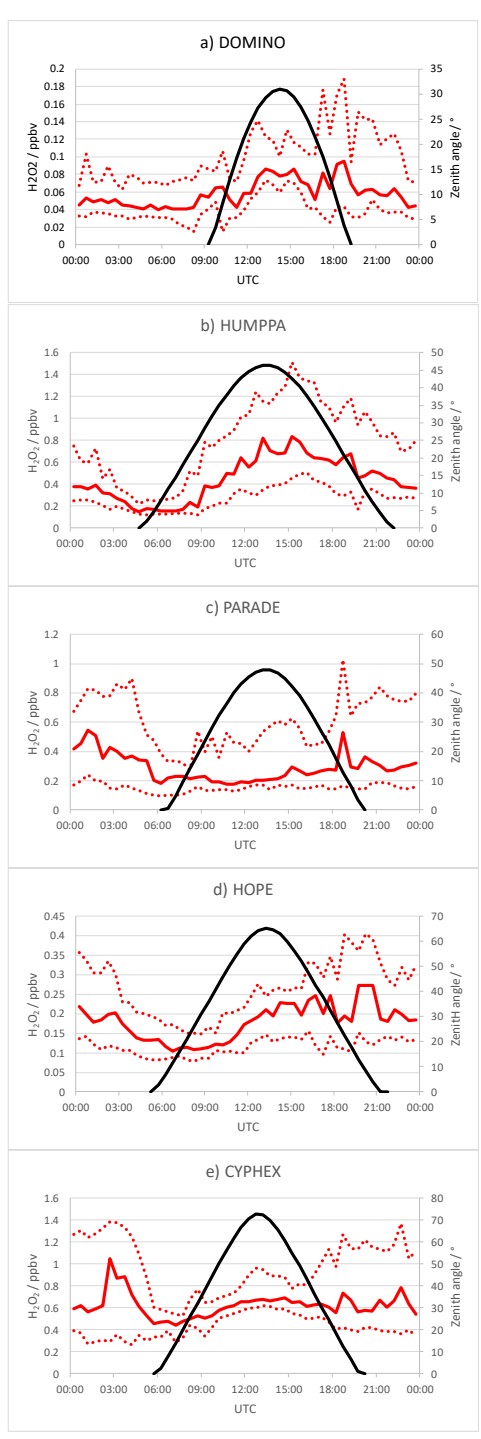

Figure 2: Hydrogen peroxide diel variation of median mixing ratios (solid read line) and 25 and 75% quartiles (dashed red lines) for 30 min bins obtained for a) DOMINO, b) HUMPPA, c) PARADE, d) HOPE and e) CYPHEX. Solar zenith angle is shown in black.



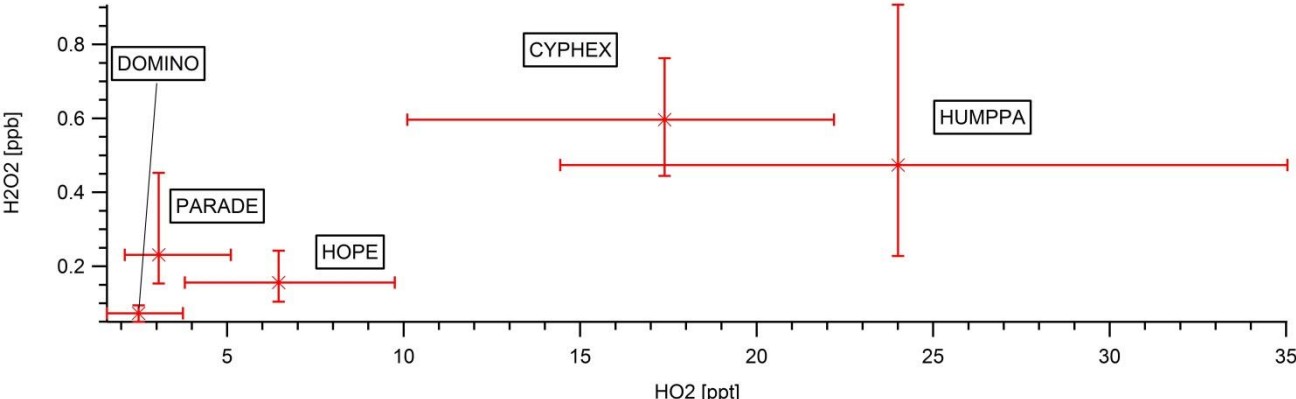

Figure 3: Relationship between $H_2O_2$ and $HO_2$ for the five campaigns. Note that only daytime values ($JNO_2 > 10^{-3}$ s$^{-1}$) have been used for the calculation of the median values and the 25 – 75 % quartiles.

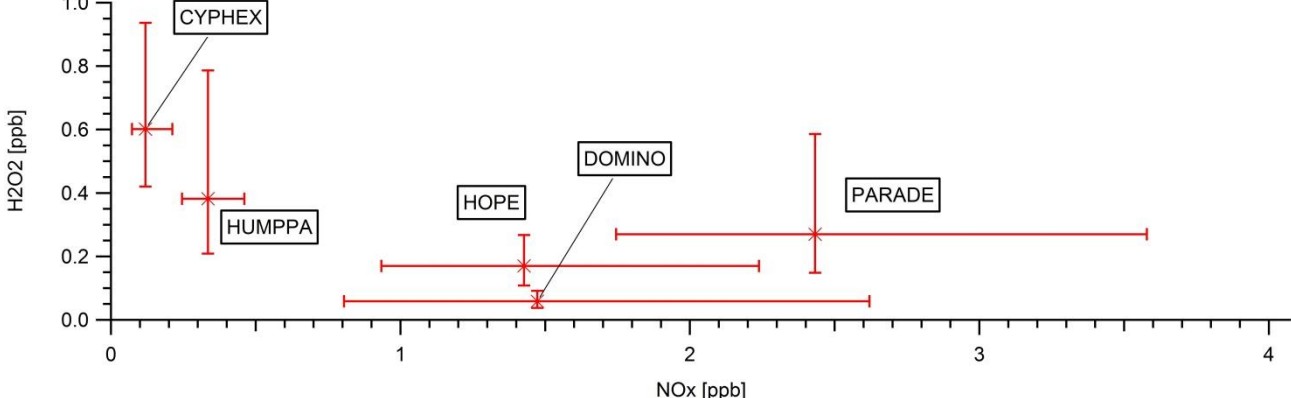

Figure 4: Relationship between $H_2O_2$ and $NOx$ for the five campaigns. Note that all data (day and night) have been used for the calculation of the median values and the 25 – 75 % quartiles.



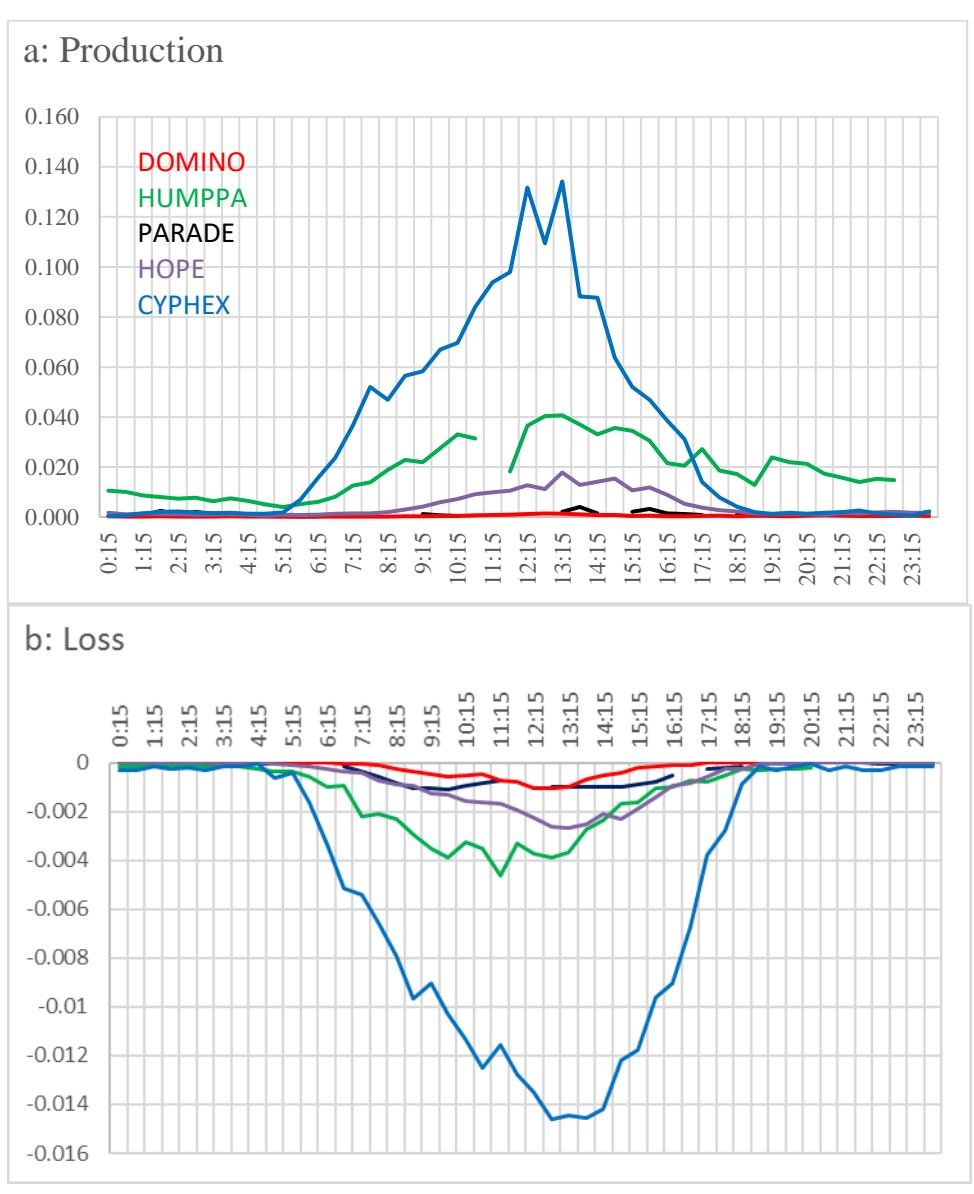

Figure 5: Photochemical production and loss of $H_2O_2$ in pptv s$^{-1}$. Note that the time used is local time.





5    Figure 6: Budget of the $H_2O$ change from the sunrise to midday for the individual campaigns. Trends, net production, deposition and transport are given in pptv h[-1]. Note the scale change from DOMINO, PARADE and HOPE (upper panel) to HUMPPA and CYPHEX (lower panel).