# Peer review of "Diurnal variability, photochemical production and loss processes of hydrogen peroxide in the boundary layer over Europe"

_Atmospheric Chemistry and Physics, 2018_

## Referee Comment (RC1) · Anonymous Referee #1 · 27 Dec 2018

The manuscript describes the diurnal variability of hydrogen peroxide and its precursors presented as median values averaged for the whole campaign for five different locations in Europe spanning southerly (Spain and Cyprus) to northerly (Finland) field sites. The median diurnal variability is proposed to be a robust signature representative for a 'typical' pattern during the campaigns.

Based on these data hydrogen peroxide budgets are calculated from photochemical production, photochemical loss and dry deposition loss. Photochemical production is linked to the main $H_2O_2$ production pathway via recombination of $HO_2$ radicals, chemical loss through reaction with OH radicals and photolysis while dry deposition is

deduced from periods with constant mixing rations but still measurable production. Dry deposition losses than should be equal to production. HO2 and OH radicals were measured as well during the campaigns. As the production of H2O2 is a fast reaction from atmospheric radicals it's production term should thus basically follow the HO2 mixing ratios. The procedure to derive dry deposition loss rates requires also a constant mixed boundary layer as well as low impact from horizontal transport with probably different composition.

General comments

The manuscript is rather difficult to read even for somebody who is familiar with atmospheric chemistry. While the chemistry introduction is state of the art, see also Tremmel et al (1993) (there also vertical profiles of H2O2 in the planetary boundary layer and free troposphere) the description of the meteorological parameters controlling the composition of air masses investigated is marginal. Only for the HUMPPA campaign in Finland a more detailed meteorological description is available. However, it's necessary to consult an additional paper. This paper includes also the vertical structure of the atmosphere which is important for both, the production term of H2O2 during the morning hours between sunrise and noon as well as for the afternoon hours deposition calculation.

Meteorological data given in the companion paper for CYPHEX are marginal. For the campaign HOPE, that's especially low in H2O2 mixing ratios no measurements of the MBL and no meteorological data are available at all.

The diurnal patterns presented are only contained in the supplement. Besides varying vertical axis units the time axis is plotted as UTC. This is basically a good way to plot a diurnal cycle, however, given the varying local time it makes a comparison of the different campaign data more difficult.

As three of the five stations (PARADE, HOPE and CYPHEX) are located either on mountain tops or in hilly terrain it is not clear, whether the assumptions made about

a single vertical column over the filed site without only marginal impact of additional horizontal transport and depth of the nocturnal boundary layer are valid. These field sites are during the day subject to significant upslope winds and even in low elevation above the site horizontal wind speeds may increase strongly. Also the nocturnal inversion layer is often far below the elevation of the field site. This is addressed in the manuscript, but it's significance is not discussed.

The data base is better for the HUMPPA campaign in Finland, however, the meteorological description of the campaign by Williams et al (2011) indicates that the summer 2010 was extraordinary hot in Finland and not representative for a 'typical' summer, making the results for HUMPPA less comparable to the other campaigns.

In summary I would recommend to consider publication after major revisions including a detailed meteorological chapter and a clear argumentation that even at the mountain stations the procedures to derive production and loss are valid. Looking at figure 2, it's obvious that hydrogen peroxide mixing ratios in Cyprus and at the Hohenpeißenberg are clearly out of phase to solar radiation and probably horizontal advection plays a major role although the chemistry is rather fast.

What is the time scale of the horizontal advection of the marine airmasses mentionded on page 13, compare to the time scale of advection of air masses at other mountain sites?

Missing mixing height data for the day and the nocturnal inversion can be obtained for example from HYSPLIT. They agree relatively well with the HUMPPA measurements. Contained also in HYSPLIT is the information of rain during the transport. This is important for example for Föhn conditions where H2O2 mixing ratios are reduced due to washout shortly before arrival at the HPB observatory.

Tremmel, H.G., Junkermann, W. Slemr, F., and Platt, U., The Distribution of Hydrogen Peroxide in the Lower Troposphere over the Northeastern U.S. during Late Summer 1988, Journal of Geophysical Research, Vol. 98, 1083-1099, 1993

Minor comments

A statement about the detection limit of the method would be helpful, AERO-LASER claims < 100 ppt, but without mentioning whether this is 1 or 3 sigma. DOMINO, PARADE and HOPE mixing rations are often very close to this level.

The argument, that the mixing layer depth cannot be used for the CYPHEX campaign on page 11, line 5-6 also holds for the HOPE campaign.

The figures in the supplement are hardly readable. The paper is not understandably without these supplementary figures.

A figure illustrating graphically the budget calculations would be helpful.

Typing errors

Page 5, lines 30 /31, Meteorologie Consult instead of Metorologie Consult

Page 11, line 27 and 29. With an uncertainty of +- 100 % it's unreasonable to estimate a deposition velocity within the percent accuracy.

Page 12. Line 32 morning instead of mourning

1. Does the paper address relevant scientific questions within the scope of ACP? YES

2. Does the paper present novel concepts, ideas, tools, or data? YES

3. Are substantial conclusions reached? NOT ALWAYS CLEAR, see detailed Review

4. Are the scientific methods and assumptions valid and clearly outlined? NOT AL-WAYS, see detailed review

5. Are the results sufficient to support the interpretations and conclusions? NOT AL-WAYS see detailed review

6. Is the description of experiments and calculations sufficiently complete and precise to allow their reproduction by fellow scientists (traceability of results)? NO

7. Do the authors give proper credit to related work and clearly indicate their own new/original contribution? YES

8. Does the title clearly reflect the contents of the paper? YES

9. Does the abstract provide a concise and complete summary? YES

10. Is the overall presentation well structured and clear? NO, see detailed review

11. Is the language fluent and precise? YES

12. Are mathematical formulae, symbols, abbreviations, and units correctly defined and used? YES

13. Should any parts of the paper (text, formulae, figures, tables) be clarified, reduced, combined, or eliminated? YES, see detailed review

14. Are the number and quality of references appropriate? YES

15. Is the amount and quality of supplementary material appropriate? NO, see detailed review

---

## Referee Comment (RC2) · Anonymous Referee #2 · 31 Mar 2019

Summary:

This study seeks to construct budgets of hydrogen peroxide (H2O2) in the planetary boundary layer based on in situ observations at five surface sites throughout Europe. The sites represent a diverse range of elevations, latitudes, and biomes, with four observational periods in the summer and one in the winter. Half-hourly binned observations at each site are classified by campaign-wide medians and 25th/75th percentile windows, to represent typical conditions and variabilities without being skewed (as the mean would be) by outliers such as rainy periods and measurements below detection limits.

[Figure]

Based on these median values, and particularly periods of increasing (morning) and steady (midday) median values, the authors construct a budget of H2O2 and diagnose the relative importance of photochemical production/loss, deposition, and transport at each of the sites. First, they use measured HO2, OH, and J(H2O2) to estimate rates of photochemical production and destruction. Depositional losses are estimated in two ways: first by assuming all H2O2 loss at night is due to deposition, and second by assuming that net photochemical production is balanced by deposition during midday hours when d[H2O2]/dt $\sim$ 0. Resulting daytime estimates are substantially higher than those derived from nighttime H2O2 loss. Finally, all d[H2O2]/dt not attributable to net photochemical production and deposition is attributed to transport. At four of the five sites, morning photochemical production is approximately balanced by deposition; the contribution of transport is therefore approximately equal to the total morning H2O2 increase.

General comments:

The authors propagate errors and uncertainties throughout the paper, but do not go on to discuss what this error means, e.g. how certain we can be (probabilistically) of the conclusions they come to about the relative importance of photochemical production/loss, deposition, and transport, and how variable these contributions are on a day-to-day basis. It would be helpful in the discussion to extend the brief description of uncertainties that focuses on HUMPPA to a wider scope, and especially to add error bars and/or daily variability to Figure 6. There are also a number of places throughout the manuscript where potential confounding factors and limiting assumptions are listed (e.g. not accounting for alkene ozonolysis, assumptions of photostationary steady state at midday) and the validity of these assumptions or the potential biases introduced are not described quantitatively, which makes it difficult to assess the total potential error from all sources in these analyses. More detail on specific occurrences of this are listed below.

A number of other concerns about the methods and their descriptions within this

manuscript are provided below within the specific comments. Most notably, I think more discussion of the use of median values and 25th/75th percentiles for the entirety of the analysis is needed. While it is clear that using medians instead of means avoids some difficulties associated with outlier values, it is not clear that the day-to-day variability can be ignored when calculating photochemical production and loss, or that this is particularly useful when the calculations could just as well be performed on un-averaged data. It would help to provide some analysis of how the calculations might change if they were not performed exclusively on campaign-wide medians. Additionally, some aspects of the calculations performed herein are not entirely clear, especially on the deposition estimates, where two complementary methods are used but the descriptions of each are intertwined. Finally, the figures could use substantial clarification; conversion of UTC to local time would help, axis titles should be added to Figure 5, axes on Figure 3 should go to zero, consistent color-coding between Figures 3-5 would be nice, and error bars should be included on Figure 6 (as well as 25/75 percentile ranges on Figure 5).

Specific comments:

P2/L11: Why does the rate coefficient depend on water vapour concentration separately from pressure?

P2/L11: Does "in general" signify that this positive dependence isn't always the case? Why not?

P2/L23: Reaction 3 does not appear to recycle the HOx from H2O2; it produces only one equivalent on HO2, and thus a cycle with R1 results in the loss of both 1OH and 1HO2.

P3/L33: Not a big deal, but the comparison with literature values seems to be spread through sections 3 and 4.

P5/L6: What does the 10-90% represent?

P5/L23-27: How are these ranges and interferences (10-30% conversion, 12-15% RO2 interference) taken into account when calculating the HO2 and the uncertainty? Do you assume a constant fraction of the HO2 to be RO2?

P6/L5: Subscript x on NOx

P6/L6: Parentheses around Fig. S1-S5

P6/L15 and elsewhere: I don't think it helps at all to use UTC instead of local time. It requires an extra step of thinking for the reader without adding any particularly useful opportunity for comparison between the campaigns. I would recommend converting everything to local time for clarity's sake.

P7/L6: Why the specific cutoff of J(NO2) > 10^(-3) s^(-1) for daytime conditions? Are the results quantitatively sensitive to the choice of cutoff?

P7/L12: "where" instead of "were"

P8/L14-15: Visual inspection does not suggest linearity; at best I think this can be described as a visible positive dependence. I also find it misleading and unhelpful to use the uncorrected HUMPPA point in the figure, and to describe the linear regression as a better fit than the initial quadratic fit, when you go on to base your analysis on the corrected value and the resulting (better) quadratic fit. We already know from your discussion of R1 above that a quadratic dependence would be expected (not taking into account other processes and dependencies). But it's also confusing to apply this analysis to overall daytime medians; the direct quadratic dependence would really only be expected on the timescales of photochemical production of H2O2. How different would this analysis be if the quadratic fit were imposed on the un-averaged data from each campaign?

P8/L23: If the HUMPPA-corrected quadratic fit provides the best correlation, why is it not shown?

P8/L24-25: I think this requires further explanation as to why you expect this quadratic

relationship to hold across environments with very different transport and deposition patterns and for median daytime values rather than instantaneous measurements of HO2 and H2O2. Figure 5 suggests that deposition and transport are highly variable and important for these locations.

P8/L26-27: If the biogenic VOCs were still quantified at DOMINO, why not correct for them in the same way even if they were low?

P8/L30-32: Why are nighttime data included in this analysis? If the results are the same either way, it would be better to at least be consistent between the two figures. What if you used day and night for the H2O2 vs HO2 analysis?

P9/L12-13: "Median values and 25 and 75 % percentiles do not include such events" - does this mean the points with rain or clouds were screened out entirely, or just that they always fall below the 25th percentile threshold? If they were screened, how so? Is this where the J(NO2) comes into play?

P9/L32-33: The implication above was that RO2 interferences had been corrected for; does this imply that there might be additional interferences?

P10/L6: In the subsequent lines, this appears to be a factor of 15, not 10

P10L10: Percent contributions of OH and photolytic losses would be more helpful here than "the same order of magnitude"

P10/L10-13: This sentence has two verbs in one clause ("is"/"yields") - either missing a conjunction or remove the "is"

P10/L14: You define NPR three times, which are also the only times you use it aside from in a table. It is probably not necessary as an acronym.

P10/L14-15: It would be nice to see the NPR as part of a figure. Figure 5 could potentially include NPR on the production panel. It should also have axis titles, and might be improved with log-scale y axes to better distinguish the shapes of the curves

with smaller magnitudes.

P10/L17: The next couple paragraphs appear to go back and forth between which method is being described (day vs. night) in a very confusing manner. I think the differentiation within the paragraphs either needs to be a lot clearer or they should be separated entirely. E.g. In the first sentence on P11, when table 2 is mentioned, it's not clarified that it's just night.

P10/L19-21: The limitations mentioned here of estimating dry deposition at night and extrapolating to the day seem like major potential sources of bias. Can you provide any estimate of the extent to which this method might underestimate deposition? Were there vertical wind speed measurements?

P10/L23: Similarly, it would be useful to estimate the extent to which this assumption of a fully established daytime boundary layer is safe or might cause bias

P10/L24-27: This is not clear. You're focusing on times when $dH2O2/dt$ is near zero, and then calculating $k(d)$ from $dH2O2/dt$, despite saying that NPR and horizontal advection also contribute. Are you subtracting those terms off? Or does this only apply to night?

P11/L11: Again, do you have any estimate of how substantial this source of error (the neglect of alkene ozonolysis in your analysis) might be?

P11/L13: Compared to the literature values for dry deposition that you go on to list, yours are much lower. What insight do we get from this?

P11/L20-23: The assumptions described here (photostationary steady state and the balance between NPR and dry deposition) require more discussion of their validity and potential introduction of uncertainty/error/bias. Do you have any estimate of what role horizontal advection might play, if air masses are coming from somewhere with different chemical characteristics?

P12/L1-2: It seems from your discussion of the differences between night and day

deposition characteristics that using the noontime values would be a more realistic substitute for the morning than linear interpolation between night and day, considering that the morning will have substantial vertical mixing. How different an answer would you get if you just use the noontime values?

P12/L12-13: These percent uncertainties should be added to the discussion below and to Figure 6.

P13/L20-23: More detail on the variability of this analysis would be useful. Is the model always 50% lower, or does it fluctuate? Does the variance in the modeled HO2 match that of the measured HO2?

P13/L24: It's my understanding from the previous sentence that the "good agreement" is largely because the measurements are corrected with the modeled RO2 to match the modeled HO2.

P13/L29-31: Do you have the necessary measurements to correct for this, or at least to weigh in on how much of a difference it makes, across your campaigns? It seems like this 33% increase (0.6 to 0.8) when considering terpene ozonolysis isn't necessarily negligible, especially if it influences the shape of the diurnal profile.

P14/L4: "mourning" should be "morning"

P14/L18: "similar" should be "similarly"

P15/L17: This statement that "the early morning rise if H2O2 mixing ratios is significantly influenced by dynamical processes" seems central to your conclusions, but given that it is based on prior estimates of net photochemical production and deposition with high uncertainty, it's not clear to what extent this statement can be supported within error estimates. What are the uncertainties on the subsequent numbers reported for each campaign? Error bars on Figure 6 would also help with this.

P15/L18-19: The sentence starting "Smaller contributions..." is missing a preposition

---

## Author Response (AR1)

**Point-by-point response to referee comments (bold) and author's changes in manuscript (red)**

**Referee 1**

The manuscript describes the diurnal variability of hydrogen peroxide and its precursors presented as median values averaged for the whole campaign for five different locations in Europe spanning southerly (Spain and Cyprus) to northerly (Finland) field sites. The median diurnal variability is proposed to be a robust signature representative for a 'typical' pattern during the campaigns.

Based on these data hydrogen peroxide budgets are calculated from photochemical production, photochemical loss and dry deposition loss. Photochemical production is linked to the main H2O2 production pathway via recombination of HO2 radicals, chemical loss through reaction with OH radicals and photolysis while dry deposition is deduced from periods with constant mixing rations but still measurable production. Dry deposition losses than should be equal to production. HO2 and OH radicals were measured as well during the campaigns. As the production of H2O2 is a fast reaction from atmospheric radicals it's production term should thus basically follow the HO2 mixing ratios. The procedure to derive dry deposition loss rates requires also a constant mixed boundary layer as well as low impact from horizontal transport with probably different composition.

General comments

The manuscript is rather difficult to read even for somebody who is familiar with atmospheric chemistry. While the chemistry introduction is state of the art, see also Tremmel et al (1993) (there also vertical profiles of H2O2 in the planetary boundary layer and free troposphere) the description of the meteorological parameters controlling the composition of air masses investigated is marginal. Only for the HUMPPA campaign in Finland a more detailed meteorological description is available. However, it's necessary to consult an additional paper. This paper includes also the vertical structure of the atmosphere which is important for both, the production term of H2O2 during the morning hours between sunrise and noon as well as for the afternoon hours deposition calculation.

**We regret that the referee considers the paper hard to read, and plan to do a better job to outline the purpose of the paper properly. Please note that we do not intend to model the H2O2 mixing ratios at different sites. This has been done e.g. in Crowley et al. (2018) using a box model for the HUMPPA campaign. Instead, we analyze the rate change of H2O2 mixing ratios in the early morning and during the night, using measured mixing ratios of precursors and photolysis rates associated with fast local photochemistry. This analysis only marginally depends on the local meteorology (with respect to temperature, pressure, relative humidity or wind speed). We agree with the referee though that the absolute level of H2O2 will strongly depend on the air mass history and thus on the synoptic meteorology. However, we feel that such an analysis is beyond the scope of our study. The relevant meteorological information necessary to evaluate Eq. 1, in particular the variation in boundary layer height, is presented in the paper, and we will scrutinize if this is well-balanced among the different campaign**

**descriptions. Information in particular on air mass history, however, is not necessary to calculate local net production rates of H2O2.**

Meteorological data given in the companion paper for CYPHEX are marginal. For the campaign HOPE, that's especially low in H2O2 mixing ratios no measurements of the MBL and no meteorological data are available at all.

**More details on the meteorology for CYPHEX can be found in Meusel et al., 2016 (doi:10.5194/acp-16-14475-2016) or Hüser et al., 2017 (doi:10.5194/acp-17-10955-2017). Since no publication on HOPE meteorology has been published so far, we will summarize the meteorology for the different campaigns in a revised version of the manuscript by including this information into Chapter 2.1 (Campaigns, observations sites and meteorology).**

**Section 2.1 Campaigns and observation site**

[revised manuscript text omitted]

The diurnal patterns presented are only contained in the supplement. Besides varying vertical axis units the time axis is plotted as UTC. This is basically a good way to plot a diurnal cycle, however, given the varying local time it makes a comparison of the

different campaign data more difficult.

**In order to compare different diurnal cycles, we included either the local solar zenith angle (Figure 2) or the JNO2 photolysis rate (supplement). We consider this more accurate than using local time, which is sensitive to the season and generally does not follows local insolation that drives the chemistry.**

Throughout the manuscript solar zenith angle has been changed to solar elevation angle

As three of the five stations (PARADE, HOPE and CYPHEX) are located either on mountain tops or in hilly terrain it is not clear, whether the assumptions made about a single vertical column over the filed site without only marginal impact of additional horizontal transport and depth of the nocturnal boundary layer are valid. These field sites are during the day subject to significant upslope winds and even in low elevation above the site horizontal wind speeds may increase strongly. Also the nocturnal in- version layer is often far below the elevation of the field site. This is addressed in the manuscript, but it's significance is not discussed.

**As mentioned above, here we only calculate the rate of change of H2O2 during the morning hours based on local sources and sinks, i.e. photochemistry and dry deposition. Based on the local production and loss, we deduce the amount of transported hydrogen peroxide that is necessary to explain the observed increase in H2O2. Transport according to Eq. 1 includes both vertical and horizontal advection. Nevertheless, considering that a regular pattern with increasing mixing ratios from sunrise to noon is observed at all cites, it is more likely that vertical transport is responsible for the positive transport, since horizontal gradients are generally small while vertical gradients can be rather large (e.g. Klippel et al., 2011).**

Page 10, lines 9-11:
Please note that horizontal advection reflecting different airmass origins will affect the absolute values of hydrogen peroxide, while the relative increase between sunrise and noon is mainly affected by local processes.

The data base is better for the HUMPPA campaign in Finland, however, the meteoro-logical description of the campaign by Williams et al (2011) indicates that the summer 2010 was extraordinary hot in Finland and not representative for a 'typical' summer, making the results for HUMPPA less comparable to the other campaigns.

**As mentioned in Williams et al., only the first half of the campaign was exceptionally hot. Furthermore, this should not affect the comparability of HUMPPA to the other campaigns. Higher temperatures will most probably lead to higher emissions of biogenic compounds. Whether this will results in higher or lower values of HOx and thus H2O2 net production is interesting, though hard to say and beyond the scope of this study.**

In summary I would recommend to consider publication after major revisions including a detailed meteorological chapter and a clear argumentation that even at the mountain stations the procedures to derive production and loss are valid. Looking at figure 2, it's obvious that hydrogen peroxide mixing ratios in Cyprus and at the Hohenpeißenberg are clearly out of phase to solar radiation and probably horizontal advection plays a major role although the chemistry is rather fast.

**As mentioned above we do not exclude transport as a process to change H2O2 concentrations. Quite the contrary, we use fast local photochemistry and dry deposition to estimate chemical processes in order to estimate the contribution of transport. Due to the fact that the effect of transport is always positive, it is more likely that it is associated with a region were H2O2 mixing ratios are systematically higher, which points to a dominant contribution from vertical transport.**

What is the time scale of the horizontal advection of the marine airmasses mentioned on page 13, compare to the time scale of advection of air masses at other mountain sites?

**This depends on the horizontal wind speed, which is comparable (2 – 6 m/s) at all mountain sites.**

Missing mixing height data for the day and the nocturnal inversion can be obtained for example from HYSPLIT. They agree relatively well with the HUMPPA measurements. Contained also in HYSPLIT is the information of rain during the transport. This is important for example for Föhn conditions where H2O2 mixing ratios are reduced due to washout shortly before arrival at the HPB observatory.

**As mentioned above the air mass history will mainly affect the absolute values of the H2O2 mixing ratio and to a much lesser extent the rate of change. Since the analysis performed here is based on observed species, the advantage of using additional HYSPLIT data is marginal and will not help our production/loss analysis.**

Tremmel, H.G., Junkermann, W. Slemr, F., and Platt, U., The Distribution of Hydrogen Peroxide in the Lower Troposphere over the Northeastern U.S. during Late Summer 1988, Journal of Geophysical Research, Vol. 98, 1083-1099, 1993

Minor comments

A statement about the detection limit of the method would be helpful, AERO-LASER claims < 100 ppt, but without mentioning whether this is 1 or 3 sigma. DOMINO, PARADE and HOPE mixing rations are often very close to this level.

**The detection limit of the instrument is mentioned on page 5 line 6 as being of the order of 25 pptv (1 sigma). It is determined from the reproducibility of zero air measurements during the individual campaigns.**

The argument, that the mixing layer depth cannot be used for the CYPHEX campaign on page 11, line 5-6 also holds for the HOPE campaign.

**We may miss the point of the referee. While nocturnal H2O2 mixing ratios show no systematic decrease during CYPHEX, a clear negative trend can be identified during HOPE indicating a first order loss process.**

The figures in the supplement are hardly readable. The paper is not understandably without these supplementary figures.

**In a revised version of the manuscript we will provide larger and clearer figures.**

The figures in the supplement have been revised.

A figure illustrating graphically the budget calculations would be helpful.

**Results of the budget calculations are summarized in Figure 6.**

Typing errors

Page 5, lines 30 /31, Meteorologie Consult instead of Metorologie Consult

Page 11, line 27 and 29. With an uncertainty of +- 100 % it's unreasonable to estimate a deposition velocity within the percent accuracy.

Page 12. Line 32 morning instead of mourning

**We will correct the typing errors in a revised manuscript.**

**The typing errors have been corrected throughout the manuscript.**

**Referee 2:**

Summary:

This study seeks to construct budgets of hydrogen peroxide (H2O2) in the planetary boundary layer based on in situ observations at five surface sites throughout Europe. The sites represent a diverse range of elevations, latitudes, and biomes, with four observational periods in the summer and one in the winter. Half-hourly binned observations at each site are classified by campaign-wide medians and 25th/75th percentile windows, to represent typical conditions and variabilities without being skewed (as the mean would be) by outliers such as rainy periods and measurements below detection limits.

Based on these median values, and particularly periods of increasing (morning) and steady (midday) median values, the authors construct a budget of H2O2 and diagnose the relative importance of photochemical production/loss, deposition, and transport at each of the sites. First, they use measured HO2, OH, and J(H2O2) to estimate rates of photochemical production and destruction. Depositional losses are estimated in two ways: first by assuming all H2O2 loss at night is due to deposition, and second by assuming that net photochemical production is balanced by deposition during midday hours when d[H2O2]/dt  0. Resulting daytime estimates are substantially higher than those derived

from nighttime H2O2 loss. Finally, all d[H2O2]/dt not attributable to net photochemical production and deposition is attributed to transport. At four of the five sites, morning photochemical production is approximately balanced by deposition; the contribution of transport is therefore approximately equal to the total morning H2O2 increase.

General comments:

The authors propagate errors and uncertainties throughout the paper, but do not go on to discuss what this error means, e.g. how certain we can be (probabilistically) of the conclusions they come to about the relative importance of photochemical pro-duction/loss, deposition, and transport, and how variable these contributions are on a day-to-day basis. It would be helpful in the discussion to extend the brief description of uncertainties that focuses on HUMPPA to a wider scope, and especially to add error bars and/or daily variability to Figure 6. There are also a number of places throughout the manuscript where potential confounding factors and limiting assumptions are listed (e.g. not accounting for alkene ozonolysis, assumptions of photostationary steady state at midday) and the validity of these assumptions or the potential biases introduced are not described quantitatively, which makes it difficult to assess the total potential error from all sources in these analyses. More detail on specific occurrences of this are listed below.

A number of other concerns about the methods and their descriptions within this manuscript are provided below within the specific comments. Most notably, I think more discussion of the use of median values and 25th/75th percentiles for the entirety of the analysis is needed. While it is clear that using medians instead of means avoids some difficulties associated with outlier values, it is not clear that the day-to-day variability can be ignored when calculating photochemical production and loss, or that this is particularly useful when the calculations could just as well be performed on un-averaged data. It would help to provide some analysis of how the calculations might change if they were not performed exclusively on campaign-wide medians. Additionally, some aspects of the calculations performed herein are not entirely clear, especially on the deposition estimates, where two complementary methods are used but the descriptions of each are intertwined. Finally, the figures could use substantial clarification; conversion of UTC to local time would help, axis titles should be added to Figure 5, axes on Figure 3 should go to zero, consistent color-coding between Figures 3-5 would be nice, and error bars should be included on Figure 6 (as well as 25/75 percentile ranges on Figure 5).

**As indicated in Table S1 the data coverage for the different campaigns varies strongly. In particular, missing HOx values make it difficult to analyze the time series of individual campaigns. Therefore, we decided to analyze diurnal variations. This can be done by using mean values and standard deviations, which are both strongly affected by values below detection limits due to e.g. cloud presence that will reduce photochemical activity. Median and inner quartiles are less sensitive but still provide a measure of the variability, excluding extreme events. The error propagation includes measurement uncertainties and atmospheric variability, with the latter being the dominant term. Exclusion of the atmospheric variability would results in errors an order of magnitude smaller. Therefore the stated values for net-production, deposition and transport are best estimates for the median values, while the error bars reflect atmospheric variability and are thus a very conservative measure of the uncertainty.**

Page 14, lines 4-7:
Please note, that the error propagation according to Eq. 4 includes measurement uncertainties and atmospheric variability, with the latter being the dominant term. Exclusion of the atmospheric variability would result in much smaller errors. Therefore, the stated values for net production, deposition and transport are best estimates for the median values, while the error bars reflect atmospheric variability and are thus a very conservative measure of the uncertainty.

Specific comments:

P2/L11: Why does the rate coefficient depend on water vapour concentration separately from pressure?

**The formation of H2O2 by the self-reaction of HO2 is strongly promoted by water vapor, which – as mentioned in the manuscript – can significantly enhance the production rate (see e.g. the data sheet for the reaction of HO2 + HO2 from IUPAC (iupac.pole-ether.fr)).**

Page 2, lines 12-14:

Note that the rate coefficient for R1 increases with increasing pressure (due to its dependence on M) and water vapour concentration [$H_2O$] (Atkinson et al., 2004; http://iupac.pole-ether.fr).

P2/L11: Does "in general" signify that this positive dependence isn't always the case? Why not?

**We erased the phrase.**

P2/L23: Reaction 3 does not appear to recycle the HOx from H2O2; it produces only one equivalent on HO2, and thus a cycle with R1 results in the loss of both 1OH and 1HO2.

**We changed the manuscript to "partly recycling HOx"**

Page 2, line 22:

Photochemical loss of $H_2O_2$ is due to either reaction with OH (R3) or photolysis (R4), partly reforming HO$_x$ radicals:

P3/L33: Not a big deal, but the comparison with literature values seems to be spread through sections 3 and 4.

**The literature review in section 3 lists previous H2O2 measurements in Europe comparing them to mixing ratio levels for this trace gas. The comparison to the literature in section 4 (discussion) deals with a comparison to other budget calculations.**

P5/L6: What does the 10-90% represent?
**This is a measure of the time resolution of the measurements. It is determined from calibration increasing from 10 % to 90 % of the signal.**

P5/L23-27: How are these ranges and interferences (10-30% conversion, 12-15% RO2 interference) taken into account when calculating the HO2 and the uncertainty? Do you assume a constant fraction of the HO2 to be RO2?

**As discussed in the manuscript the early HO2 measurements (100% conversion) suffered from an unquantified RO2 interference, mainly caused by biogenic alkenes. For HUMPPA 2010, which was conducted in a boreal forest, Hens et al., 2014 estimated an uncertainty of 30 % of the HO2 data due to the uncorrected RO2 interference. Crowley et al., 2018 found in a constrained box model study of the HUMPPA dataset 30% interference during noon by RO2, confirming the finding by Hens et al. but estimated for the nighttime and early morning hours a contribution of the RO2 interference of up to 100 % to the HO2 signal. In this study, we consider for the calculating the H2O2 production rate during HUMPPA a weighted all day interference of 50 % on the HO2 dataset. As discussed in section 4 around noon this leads to HO2 mixing ratios that are approx. 10 % lower than simulated values. The consequence of this uncertainty is discussed in section 4.**

Page 6, lines 16-20:

Crowley et al. (2018) found in a data-constrained box model study that during HUMPPA at noon 30 % occurred due to $RO_2$, confirming the finding by Hens et al. (2014). For the early morning and nighttime hours $RO_2$ interference was significantly larger. During later campaigns (PARADE, HOPE, CYPHEX) the reduction of the amount of NO used to convert $HO_2$ to OH resulted in $RO_2$ interferences of the order 12 to 15 % (Mallik et al., 2018).

Page 9, lines 9-13:

In a recent modelling study, Crowley et al. (2018) determined the contribution of $RO_2$ to the measured $HO_2$ during the daylight hours to be of the order of 30 % around noon and close to 100 % around sunrise and sunset. If we correct the HUMPPA data by a weighted all day value of 50 % for this potential interference, median daytime $HO_2$ is reduced from 24 pptv (14 to 35 pptv for the 25 – 75 % percentiles) to 12 pptv (7 – 17.5 pptv).

P6/L5: Subscript x on NOx

P6/L6: Parentheses around Fig. S1-S5

**Changed.**

P6/L15 and elsewhere: I don't think it helps at all to use UTC instead of local time. It requires an extra step of thinking for the reader without adding any particularly useful opportunity for comparison between the campaigns. I would recommend converting everything to local time for clarity's sake.

**Since local time depends on time zone and season and is only weakly related to the solar cycles, we prefer to stick to UTC. Solar zenith angles in Fig.2 and JNO2 in the supplement should make it easy to identify sunrise, noon and sunset.**

P7/L6: Why the specific cutoff of J(NO2) > 10ˆ(-3) sˆ(-1) for daytime conditions? Are the results quantitatively sensitive to the choice of cutoff?

**This is an often used cutoff value to differentiate between night and day.**

P7/L12: "where" instead of "were"

**Changed**

P8/L24-25: I think this requires further explanation as to why you expect this quadratic relationship to hold across environments with very different transport and deposition patterns and for median daytime values rather than instantaneous measurements of HO2

and H2O2. Figure 5 suggests that deposition and transport are highly variable and important for these locations.

**The mixing ratio of H2O2 at a given location depends on transport and local photochemistry. Since the production term is proportional to the square of HO2, and the H2O2 can be expected to be dependent on its production rate, one would expect to find such a relationship.**

P8/L26-27: If the biogenic VOCs were still quantified at DOMINO, why not correct for them in the same way even if they were low?

**As mentioned above, a correction for the RO2 bias of the HO2 measurements for DOMINO and HUMPPA is not possible, but instead the effect is summarized through a larger uncertainty. It is expected that this has a stronger effect during HUMPPA due to generally higher alkene concentration. During DOMINO alkene concentrations are much smaller and thus the HO2 mixing ratios are less likely be affected by such an interference.**

P8/L30-32: Why are nighttime data included in this analysis? If the results are the same either way, it would be better to at least be consistent between the two figures. What if you used day and night for the H2O2 vs HO2 analysis?

**As mentioned above HO2 during the night is zero, while this is not the case for NOx (as is H2O2). Limiting the plot to daytime values does not change the relation.**

P9/L12-13: "Median values and 25 and 75 % percentiles do not include such events"
- does this mean the points with rain or clouds were screened out entirely, or just that they always fall below the 25th percentile threshold? If they were screened, how so? Is this where the J(NO2) comes into play?

**The data were not filtered for rain or cloud effects and the JNO2 filter was only used to differentiate between night and day. Under cloudy conditions or rain the photochemical activity is strongly reduced due to less insolation, which results in lower radical levels and thus H2O2. Since cloud/rain periods were rare during the campaigns they fall indeed below the 25th percentile threshold.**

P9/L32-33: The implication above was that RO2 interferences had been corrected for; does this imply that there might be additional interferences?

**According to the model results of Crowley et al (2018), the contribution of the interference by RO$_2$ on the HO$_2$ signal was much greater during nighttime than during the day, which results from the large (modelled) organic peroxy radical mixing ratios at night compared to the (modelled) HO$_2$ at night. Hence, the nighttime production of H$_2$O$_2$ during HUMPPA is likely to be an artefact, as mentioned. We shall extend the text to clarify this.**

Page 6, lines 16-20:

Crowley et al. (2018) found in a data-constrained box model study that during HUMPPA at noon 30 % occurred due to $RO_2$, confirming the finding by Hens et al. (2014). For the early morning and nighttime hours $RO_2$ interference was significantly larger. During later campaigns (PARADE, HOPE, CYPHEX) the reduction of the amount of NO used to convert $HO_2$ to OH resulted in $RO_2$ interferences of the order 12 to 15 % (Mallik et al., 2018).

Page 9, lines 9-13:

In a recent modelling study, Crowley et al. (2018) determined the contribution of $RO_2$ to the measured $HO_2$ during the daylight hours to be of the order of 30 % around noon and close to 100 % around sunrise and sunset. If we correct the HUMPPA data by a weighted all day value of 50 % for this potential interference, median daytime $HO_2$ is reduced from 24 pptv (14 to 35 pptv for the 25 – 75 % percentiles) to 12 pptv (7 – 17.5 pptv).

P10/L6: In the subsequent lines, this appears to be a factor of 15, not 10

**Changed to “by an order of magnitude”**

P10L10: Percent contributions of OH and photolytic losses would be more helpful here than "the same order of magnitude"

**Changed to “are of similar magnitude”**

P10/L10-13:  This sentence has two verbs in one clause ("is"/"yields") - either missing  a conjunction or remove the "is"

**Changed**

P10/L14: You define NPR three times, which are also the only times you use it aside from in a table. It is probably not necessary as an acronym.

**Changed**

P10/L14-15: It would be nice to see the NPR as part of a figure. Figure 5 could potentially include NPR on the production panel. It should also have axis titles, and might be improved with log-scale y axes to better distinguish the shapes of the curves with smaller magnitudes.

**The production term is generally an order of magnitude larger than the photochemical loss. Therefore, the net production is equivalent to the production term. We will change Figure 5 by adding axis title and we will check whether a log scale will improve the readability of the graph.**

Figure 5 has been revised by adding titles to the axis. A change to log scale has not been made, since negative values cannot be converted.

P10/L17: The next couple paragraphs appear to go back and forth between which method is being described (day vs. night) in a very confusing manner. I think the differentiation within the paragraphs either needs to be a lot clearer or they should be separated entirely. E.g. In the

first sentence on P11, when table 2 is mentioned, it's not clarified that it's just night.

P10/L19-21: The limitations mentioned here of estimating dry deposition at night and extrapolating to the day seem like major potential sources of bias. Can you provide any estimate of the extent to which this method might underestimate deposition? Were there vertical wind speed measurements?

P10/L23: Similarly, it would be useful to estimate the extent to which this assumption of a fully established daytime boundary layer is safe or might cause bias

P10/L24-27: This is not clear. You're focusing on times when dH2O2/dt is near zero, and then calculating k(d) from dH2O2/dt, despite saying that NPR and horizontal advection also contribute. Are you subtracting those terms off? Or does this only apply to night?

P11/L11: Again, do you have any estimate of how substantial this source of error (the neglect of alkene ozonolysis in your analysis) might be?

P11/L13: Compared to the literature values for dry deposition that you go on to list, yours are much lower. What insight do we get from this?

P11/L20-23: The assumptions described here (photostationary steady state and the balance between NPR and dry deposition) require more discussion of their validity and potential introduction of uncertainty/error/bias. Do you have any estimate of what role horizontal advection might play, if air masses are coming from somewhere with different chemical characteristics?

**Since the above eight comments deal with the determination of the deposition velocity, we will address them together. The dry deposition of a trace gas depends on its loss on a surface (described by a surface resistance) and transport to the surface, mainly due to turbulence. During the night, the transport is small due to low turbulence. This leads to a small deposition velocity. During the day solar irradiation forces turbulence and thus stronger transport to the surface. In addition, the leave stomata are closed during the night, but are opened during the day and reduce the surface resistance. Both effects lead to higher deposition velocities. During the morning, the deposition velocity changes from low to high values, mainly with the increase of turbulence. Therefore, we use an interpolation instead of the maximum value for noon. These interpolated values are similar to the cited values in the literature that were obtained during daylight hours. The assumption of a well-established boundary layer around noon is based on observations (e.g. Figure 4 in Ouwersloot et al., 2012). Based on aircraft observations, horizontal gradients in H2O2 mixing ratios are generally small (e.g. Klippel et al., 2011) so that we do not expect a strong influence of advection.**

**The text will be amended to remove any confusion related to how the Vdep was calculated.**

Page 11, lines 12-23:

The dry deposition of a trace gas depends on its loss at a surface (described by a surface resistance) and transport to the surface, mainly by turbulence. During the night the transport term is small due to low turbulence and thus we expect a low deposition velocity. In a first step, we therefore estimate the deposition loss rate constant from the decrease of $H_2O_2$ mixing ratios during the night, when photochemical production and loss, as well as vertical entrainment are assumed to be negligible. This estimate of the dry deposition sink is a lower limit, since it

neglect thermally driven turbulence and dry deposition due to stomatal uptake by vegetation, which occurs only during the day and does not take into account day-night changes in the rate of turbulent transport to the ground (e.g. Nguyen et al., 2015). In order to account for the contribution of enhanced turbulence and stomatal uptake, as a second step we also estimate dry deposition loss during local noontime. During this time of the day, associated with maximum $H_2O_2$ mixing ratios, it can be assumed that the daytime boundary layer is fully established and vertical intrusion is at minimum. Concentrating on periods with $d[H_2O_2]/dt \sim 0$, only net chemical production, dry deposition and horizontal advection will influence the concentration of $H_2O_2$.

P12/L12-13: These percent uncertainties should be added to the discussion below and to Figure 6.

**See above for a discussion about uncertainties and atmospheric variability.**

P13/L20-23: More detail on the variability of this analysis would be useful. Is the model always 50% lower, or does it fluctuate? Does the variance in the modeled HO2 match that of the measured HO2?

**A detailed discussion of the comparison between modelled and observed HO2 can be found in the paper by Crowley et al., 2018.**

P13/L24: It's my understanding from the previous sentence that the "good agreement" is largely because the measurements are corrected with the modeled RO2 to match the modeled HO2.

**Correct**

P13/L29-31: Do you have the necessary measurements to correct for this, or at least to weigh in on how much of a difference it makes, across your campaigns? It seems like this 33% increase (0.6 to 0.8) when considering terpene ozonolysis isn't necessarily negligible, especially if it influences the shape of the diurnal profile.

**Since we interpolate between night- and daytime deposition velocities this small absolute difference in the nighttime deposition velocity is insignificant.**

P14/L4: "mourning" should be "morning" P14/L18: "similar" should be "similarly"

**Changed.**

P15/L17: This statement that "the early morning rise if H2O2 mixing ratios is significantly influenced by dynamical processes" seems central to your conclusions, but given that it is based on prior estimates of net photochemical production and deposition with high uncertainty, it's not clear to what extent this statement can be supported within error estimates. What are the

uncertainties on the subsequent numbers reported for each campaign? Error bars on Figure 6 would also help with this.

**We will change the text to "is significantly influenced by dynamical processes".**

Page 16, lines 16-17:
Using the individual terms for $H_2O_2$ photochemical production, photochemical loss and dry deposition we could show that the early morning rise in $H_2O_2$ mixing ratios is influenced by dynamical processes.

P15/L18-19: The sentence starting "Smaller contributions..." is missing a preposition
**Changed.**